# Skin Lesion Phenotyping via Nested Multimodal Contrastive Learning

## Abstract

We introduce SLIMP (Skin Lesion Image-Metadata Pre-training) for learning rich representations of skin lesions through a novel nested contrastive learning approach that captures complementary information between images and metadata. Melanoma detection and skin lesion classification based solely on images, pose significant challenges due to large variations in imaging conditions (lighting, color, resolution, distance, etc.) and lack of clinical and phenotypical context. Clinicians typically follow a holistic approach for assessing the risk level of the patient and for deciding which lesions may be malignant and need to be excised by considering the patient's medical history as well as the appearance of other lesions of the patient. Inspired by this, SLIMP combines the appearance and the metadata of individual skin lesions with patient-level metadata relating to their medical record and other clinically relevant information. By fully exploiting all available data modalities throughout the learning process, the proposed pre-training strategy improves performance compared to other pre-training strategies on downstream skin lesion classification tasks, highlighting the learned representations quality.

## 1 Introduction

The analysis of skin lesion characteristics is an important part of dermatological examination, allowing clinicians to recognize potential skin malignancies and establish suitable follow-up actions and treatment plans. Among skin malignancies, melanoma, although having a lower incidence with respect to other skin cancers, has a significantly heavier impact on the patient health in terms of morbidity and mortality. There are over 330,000 cases of melanoma diagnosed worldwide every year, leading to more than 55,000 deaths annually (Arnold et al., 2022), with data suggesting an increased incidence in the last years (Sun et al., 2024). Importantly, when detected early (stage I-II), melanoma can be cured in the majority of cases through surgical excision. This suggests the importance of developing efficient and effective methods for early detection of melanoma and other types of skin cancers.

Numerous works in the literature have attacked the problem of classifying skin lesions based on their appearance (Hasan et al., 2023; Adegun & Viriri, 2021), largely supported by the monumental effort put forward by the International Skin Imaging Collaboration (ISIC) for constructing the ISIC datasets and organizing the corresponding challenges from 2016. In dermatological clinical practice though, clinicians do not base their decisions solely on the appearance of the patient's individual lesions, but also consider additional lesion characteristics, as well as their skin phenotype and habits. Drawing inspiration from this, recent datasets, including SLICE-3D (Kurtansky et al., 2024), typically include lesion and patient metadata (Pacheco et al., 2020; Tschandl et al., 2018; Mendonça et al., 2013).

Despite the significant effort dedicated in producing large collections of skin lesion data, the amount of annotated skin lesion data corresponding to malignant lesions still lies far from those available for other computer vision tasks, making the development of deep-learning methods that rely on large data quantities troublesome. The combination of different skin lesion datasets can alleviate these problems, yet differences in imaging modalities (clinical vs dermoscopic images) and metadata attributes pose an important challenge in their effective use for training deep-learning models. Suitable pre-text tasks offering self-supervision have proven to be invaluable in such scenarios, enabling the models to learn rich representative features that can be subsequently employed to address downstream tasks even when less data are available.

Building on these observations, we introduce SLIMP (Skin Lesion Image-Metadata Pre-training), a novel pre-training approach for skin lesions based on nested multi-modal contrastive learning, which aims to exploit all available data modalities across all stages of the learning process. SLIMP captures relations between the appearance of the lesions and the metadata associated with them in the context of the patient-level metadata. By incorporating both lesion and patient level metadata, the proposed method fully exploits information that is complementary to the appearance of the lesions, producing representative and generalizable features for skin lesions that lead to improved performance in downstream tasks. To enable effective transfer to target datasets, we employ an efficient continual pre-training approach for addressing the problems that arise from the differences that typically occur between the metadata structure and imaging modalities of different datasets. Additionally, by exploiting the structure of the common images-metadata embedding space learned during the pre-training phase, we propose an extrapolation technique for enriching datasets that do not contain metadata, by transferring metadata from a reference dataset based on their agreement with the target images.

The contributions of this work are the following:

1. We propose a multi-modal pre-training strategy based on a novel nested contrastive learning schema for producing rich skin lesion representations by leveraging metadata both at the lesion and patient levels which complement the visual information of the lesion images;

2. We adapt the learned representations on target datasets through efficient continual pre-training, effectively addressing differences in metadata attributes and imaging modalities;

3. We propose a metadata extrapolation strategy for enhancing image-only datasets using suitable reference metadata;

4. The proposed nested multi-modal pre-training strategy achieves improved performance in downstream tasks compared to competing pre-training strategies and strong baselines, including fully-supervised approaches.

## 2 RELATED WORK

**Multi-modal self-supervised representation learning** is used for enhancing image-based models by incorporating different data modalities, especially for tasks where additional context provides useful information for improved task performance. In this context, CLIP (Radford et al., 2021) introduced a method for learning image-text representations through a contrastive learning paradigm. By linking each image to a natural language description, CLIP captures subtle patterns and nuances, creating representations that can accommodate different applications. This paradigm has been followed by a large number of works, including (Zhai et al., 2023) and (Tschannen et al., 2025). In a domain-specific context, the work of Bourcier et al. (2024) adopted a multi-modal pre-training approach for learning representations based on satellite imagery and associated metadata, showing that the additional context provided by metadata leads to improved performance in downstream tasks.

Regarding contrastive learning performed across taxonomies, Zhang et al. (2022) introduced hierarchical contrastive pre-training for images, allowing to consider labels organized in a taxonomy, by proposing a natural extension of the contrastive loss for hierarchical label relations as well as a constraint enforcing loss for separating distinct lineages. Fan et al. (2024) used three levels of contrastive learning for improved sentiment analysis by incorporating various features combinations of the available data modalities.

In the medical domain, the work of Jiang et al. (2023) highlighted the importance of taking into account the patient-slide-patch hierarchy in learning suitable representations for cancer diagnosis based on whole-slide images. On the other hand, Wang et al. (2023) used a contrastive loss spanning multiple levels across the same modality, ranging from patient-level to observation-level, for maximizing information utilization of the available data, leading to stronger representations for medical time-series analysis and classification.

In this work we adopt a contrastive learning strategy across two distinct levels of metadata, modeled as one level nested within the other, as patient-level metadata are shared while lesion-level metadata regard individual skin lesions. This scheme encourages learning of more representative skin-lesion

representations that can assist in the downstream skin lesion classification task while offering improved generalization across different patients.

**Dermatology-specific representation learning** has been pursued in several approaches specifically tailored for skin lesion analysis, going beyond generic computer vision models. WhyLesionCLIP Yang et al. (2024) adapts the CLIP architecture using a supervised objective, fine-tuning it on a large corpus of skin lesion images and biomedical text descriptions to capture domain-specific semantics. Similarly, PanDerm Yan et al. (2025) leverages large-scale vision-language pre-training to address diverse dermatological tasks. Addressing the specific challenge of class imbalance in medical datasets, SBCL Hou et al. (2023) employs a supervised subclass-balancing contrastive strategy to improve representation learning on long-tailed distributions. While these methods effectively leverage unstructured text or specialized sampling strategies, SLIMP introduces a distinct paradigm by modeling the inherent structured, compositional hierarchy of tabular metadata, a critical clinical modality that offers complementary constraints to text-based or image-only approaches.

**Continual pre-training** has become a key strategy to make pretrained models more specialized and effective for real-world applications, where domain-specific knowledge is often crucial. In this context, Gururangan et al. (2020) demonstrated that simply continuing to pretrain a language model on domain-specific texts substantially improves the accuracy across diverse tasks, even when labeled data is limited. Liu et al. (2021) developed a continual pre-training framework for the mBART model to boost machine translation for low-resource languages, where translation data is often limited or nonexistent. By generating mixed-language text from available monolingual resources, they enabled mBART to 'self-train' on noisy but representative data and extend its language skills to previously unseen languages. In the domain of geospatial analysis, Mendieta et al. (2023) tackled the resource-intense needs of geospatial applications with a continual pre-training method that exploits the rich representations coming from large-scale image datasets like ImageNet-22k.The work of Reed et al. (2022) extended this adaptive pre-training to general computer vision, aiming to address the high costs of self-supervised learning. Their approach, utilize existing pretrained models as a starting point to accelerate learning, achieving improved accuracy with fewer resources.

Multi-modal continual pre-training has only recently been explored, mainly regarding the adaptation of vision-language models (Roth et al., 2024; Chen et al., 2025). In the medical domain, Ye et al. (2024) proposed continual pre-training for multi-modal medical data in a multi-stage manner to avoid interference between image and non-image modalities during learning. The proposed method makes use of continual pre-training to fully exploit target dataset metadata. Due to the differences in the recorded attributes, continual pre-training allows adapting the metadata encoder accordingly, leading to improved classification performance. To the best of our knowledge, this is the first work that explores the use of multi-modal continual pre-training for tabular metadata, allowing to fully exploit the available metadata of target domains. Importantly, the proposed continual pre-training strategy does not rely on target labels, which are not always available in the context of skin lesion classification and other similar medical applications.

**Data enhancement through retrieval** has been proposed in the natural language processing domain under different settings. In Borgeaud et al. (2022), a retrieval-enhanced language model (RETRO) is introduced augmenting a frozen language model allowing retrieval from a large text database for improving its performance. In a similar direction, Träuble et al. (2023) proposed a discrete key-value bottleneck architecture considering pairs of sparse, separable and learnable key-value codes.

The work of Norelli et al. (2023) applies the idea in a multi-modal setting, establishing image-text correspondences using independently pre-trained image and text encoders by exploiting similarities within each modality in combination with a reduced dataset of known image-text correspondences. We consider a retrieval-enhanced variant of SLIMP for allowing multimodal classification even for image-only datasets, by matching metadata from a reference dataset.

## 3 METHOD

In this section we present SLIMP, a self-supervised pre-training approach with a nested contrastive loss. Given a reference skin-lesion classification dataset providing metadata at the lesion and at the patient levels, the proposed approach aims to learn representative and generalizable skin lesion

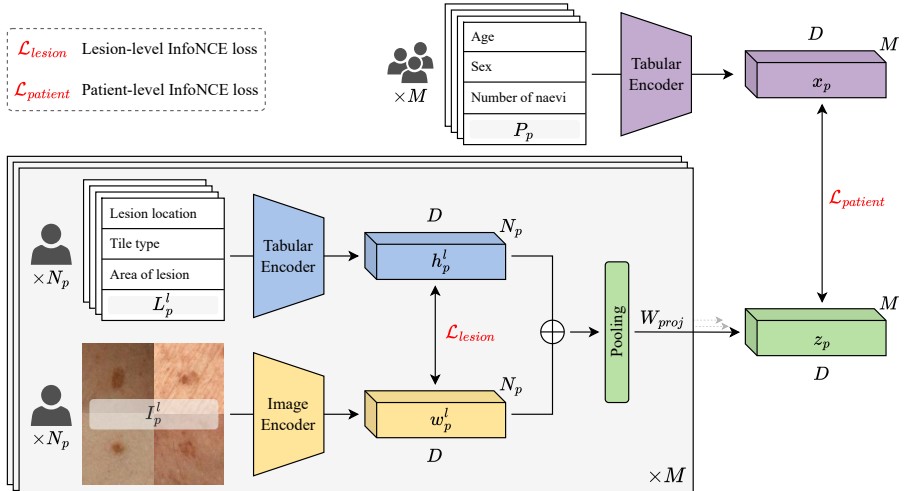

Figure 1: SLIMP architecture. An inner multi-modal contrastive loss is employed to maximize agreement among images of skin lesions and the corresponding metadata. Skin lesion image and metadata representations of a patient are aggregated, summarizing the lesion phenotype. At the patient level, agreement between the estimated lesion phenotype and the patient metadata is pursued through an outer contrastive loss.

representations by combining appearance information with information stemming from the corresponding metadata at both levels. Two strategies are then proposed for adapting these representations to target datasets in a way that fully exploits the available metadata, even when their structure and content differ from the source data. This leads to enhanced performance on downstream classification and retrieval tasks by leveraging multi-modal information about the skin lesions. The notation used throughout this section is summarized in Table 5.

## 3.1 NESTED CONTRASTIVE MULTI-MODAL LEARNING

The overall approach is presented in Figure 1 and summarized in Algorithm 1. For each patient, $p \in \{1, ..., M\}$ our model processes $N_p$ lesion images $\{I_p^l\}_{l=1}^{N_p}$ with an image encoder to extract image-based features $\{w_p^l \in \mathbb{R}^D\}_{l=1}^{N_p}$, where $D$ denotes the dimensionality of the image embedding. In parallel, the model processes the corresponding lesion-specific tabular metadata $\{L_p^l\}_{l=1}^{N_p}$ with a

---

**Algorithm 1:** SLIMP Nested Contrastive Learning Pseudocode

---

**Data:** Lesion images: $\{\{I_p^l\}_{l=1}^{N_p}\}_{p=1}^M$, lesion metadata: $\{\{L_p^l\}_{l=1}^{N_p}\}_{p=1}^M$, patient metadata: $\{P_p\}_{p=1}^M$.

Sample a batch of $B$ patients

$\mathcal{L}_{lesions} = 0$

**for** $p \in \{1, \ldots, B\}$ **do**

    Build batch of $N$ lesion image-metadata pairs from patient $p$

    **for** $l \in \{1, \ldots, N\}$ **do**

        $w_p^l$ = `ImageEncoder(`$I_p^l$`)`

        $h_p^l$ = `LesionTabularEncoder(`$TL_p^l$`)`

    **end**

    $\mathcal{L}_{lesions}$ += $\frac{1}{B}$`InfoNCELoss(`$\{w_p^l\}_{l=1}^N, \{h_p^l\}_{l=1}^N$`)`

    $z_p$ = `Linear(AvgPool(`$\{(w_p^l, h_p^l)\}_{l=1}^N$`))`

**end**

$\{x_p\}_{p=1}^B$ = `PatientTabularEncoder(`$\{TP_p\}_{p=1}^B$`)`

$\mathcal{L}_{patient}$ = `InfoNCELoss(`$\{(z_p, x_p)\}_{p=1}^B$`)`

$\mathcal{L}_{total} = \lambda \cdot \mathcal{L}_{lesions} + (1 - \lambda) \cdot \mathcal{L}_{patient}$

Figure 2: Use of learned representations for skin lesion classification. Classification of a skin lesion using corresponding data modalities (image+metadata) is shown on the left. Classification of a skin lesion image using the retrieval-based metadata extrapolation method is shown on the right.

tabular metadata encoder to extract metadata-based feature representations $\{h_p^l \in \mathbb{R}^D\}_{l=1}^{N_p}$ on a lesion level. The resulting lesion-level representations are jointly optimized using an inner loss ($\mathcal{L}_{lesions}$) based on InfoNCE (Mendieta et al., 2018) to maximize the agreement between the lesions of a single patient. By maximizing the similarity between the corresponding lesion image-metadata pairs and, analogously, minimizing the cosine similarity between non-matching pairs, the model learns multi-modal lesion-level representations. The two lesion-level modalities are merged via concatenation, which has been shown to be a simple yet effective strategy (Weng et al., 2019) for obtaining a combined lesion-level representation $\{(w_p^l, h_p^l)\}_{l=1}^{N_p}$. These combined lesion representations are aggregated for all the lesions of a patient by applying average pooling, and they are subsequently linearly transformed into a single vector $z_p \in \mathbb{R}^D$, summarizing the lesion phenotype of the patient. At the outer level, SLIMP processes the patient-specific tabular metadata ($P_p$) utilizing an outer tabular metadata encoder, yielding a representation $x_p \in \mathbb{R}^D$. An outer InfoNCE loss ($\mathcal{L}_{patient}$) is then applied between the patient-level metadata representation $x_p \in \mathbb{R}^D$ and the patient-level lesion phenotype representation $z_p \in \mathbb{R}^D$ obtained at the inner level. This nested contrastive pre-training framework enables the model to learn rich skin lesion representations that take into consideration the patient's phenotype. The complete loss formulation is provided in Section C.

## 3.2 SELF-SUPERVISED IMAGE-METADATA CONTINUAL PRE-TRAINING

Due to differences in clinical practice, regulatory context, and other related factors, skin-lesion datasets show significant variability both as far as imaging modality is concerned and because of quantitative and qualitative different behavioral and clinical attributes collected from the patients. To overcome this inherent difficulty, we propose a multi-modal continual pre-training approach for adapting the representations learned by pre-training SLIMP on a large reference dataset to potentially smaller datasets with diverging metadata and/or imaging modalities.

To achieve this unsupervised adaptation, the image and tabular encoders are fine-tuned by employing the same nested multi-modal contrastive architecture to the target domain. To address the domain differences, only the first (embedding) layers of the image and the tabular encoders are modified, keeping deeper layers frozen. This helps to preserve the structure of the common learned space, alleviating catastrophic forgetting, while allowing for the target domain data to be suitably mapped to this common space. Implementation details are provided in Section D, while we compare this strategy against fine-tuning all the model parameters in Section F. For completness, we specify that in cases where only lesion-level metadata are available, a *flattened* SLIMP variant is considered, taking into account solely the lesion images and the corresponding metadata.

## 3.3 SUPERVISED SKIN-LESION CLASSIFICATION USING MULTI-MODAL FEATURES

To assess the quality of the features learned by SLIMP, we consider the downstream skin-lesion classification task following the standard evaluation protocol in self-supervised learning literature by employing a supervised linear classifier operating on the concatenation of the features produced by the frozen image encoder and the two frozen tabular encoders that process lesion- and patient-level metadata, respectively.

**Dataset enhancement via metadata extrapolation** When the target dataset lacks metadata, a retrieval-based metadata extrapolation approach is used for artificially enhancing the target dataset by creating metadata pseudo-modalities. As lesion metadata are tightly related to the corresponding images, we consider the possibility of enhancing datasets that do not provide metadata by constructing pseudo-modalities of patient-level and lesion-level metadata using the corresponding modalities of the reference dataset on which the SLIMP model has been pre-trained. Drawing inspiration from Norelli et al. (2023) and building on the fact that the lesion- and patient-level modalities have been trained to maximize agreement, we use the encoding of the lesion images to retrieve the metadata of the original dataset that exhibit the highest similarity and use them on downstream tasks. A detailed discussion regarding the structure of the SLIMP embedding space, supporting the validity of this approach, is provided in Section E, while Section F provides an ablation.

This classification paradigm is presented in Figure 2 (right). Specifically, the model utilizes only the images $I_p^l$ from the target dataset, passing them through the image encoder of the SLIMP model that has been pre-trained on the reference dataset, providing the target dataset image representations $w_p^l$. Based on these features, a two-step metadata retrieval process is performed to incorporate additional context from the reference dataset metadata representations. First, we compare $w_p^l$ with the features $\tilde{h}^{l'}$ derived from the pre-trained SLIMP lesion metadata encoder, and we retrieve the vector $\hat{h}^{l'}$ with the highest similarity. The combined feature set $\{(w_p^l, \hat{h}^{l'})\}$ is linearly transformed into a single patient-level vector $\hat{z}_p$, which is then compared with the features $\tilde{x}_{p'}$ derived from the pre-trained SLIMP patient metadata encoder to retrieve the most relevant $\hat{x}_{p'}$. By adding pseudo-modalities on both the patient and the lesion level, this retrieval process produces three feature vectors for each image of the target dataset $\hat{y}_p^l : \{(w_p^l, \hat{h}^{l'}, \hat{x}_{p'})\}$ that can be used for lesion classification.

## 4 EXPERIMENTAL EVALUATION

### 4.1 DATASETS

Evaluation is performed considering five widely used, public skin lesion datasets, which differ in key aspects, including dataset size, imaging modality (dermoscopic or clinical), availability of metadata (such as the number of patient clinical features), and degree of class imbalance. SLICE-3D (Kurtansky et al., 2024) is used as a reference dataset, both due to the significantly higher number of samples and the richness of the metadata features. PAD-UFES-20 (Pacheco et al., 2020), HIBA (ISIC, 2024), HAM10000 (Tschandl et al., 2018), and PH2 (Mendonça et al., 2013) are considered as target datasets. The main characteristics of the datasets are summarized in Table 1, while Section B provides additional details.

### 4.2 IMPLEMENTATION

Unless otherwise stated, we employ ViT-Small (Dosovitskiy et al., 2021) as a transformer-based image encoder and TRACE (Christopoulos et al., 2025) as a transformer-based encoder for clinical tabular data. We train the model for 150 epochs on an NVIDIA RTX A6000 GPU with 48GB of VRAM. For pre-training the model on the SLICE-3D dataset, we consider a batch size of $B = 4$ patients and $N = 100$ lesions. For continual pre-training on target datasets, we fine-tune the embedding layers of the image and metadata encoders, keeping their attention layers frozen. We have observed that this strategy leads to increased performance in downstream tasks. During continual pre-training, the batch size is increased to 64 patients. For evaluating the intrinsic quality of the feature representations, we employ the standard practice established in the self-supervised learning literature, randomly splitting

Table 1: Main aspects of skin lesion datasets considered in the evaluation.

| Dataset | Image Modality | Number of Samples | Number of Patients | Targets | Patient Missing Values(%) | Lesion Missing Values(%) | Metadata |
|---|---|---|---|---|---|---|---|
| SLICE-3D | Clinical | 401,059 | 1,042 | Benign/Malignant | 1.78% | 0.04% | Patient/Lesion |
| PAD-UFES-20 | Clinical | 2,298 | 1,373 | Multiclass | 32.20% | 7.00% | Patient/Lesion |
| HIBA | Mixed | 1,616 | 623 | Multiclass | 21.20% | 12.80% | Patient/Lesion |
| HAM10000 | Dermoscopic | 10,015 | N/A | Multiclass | 0.57% | 1.17% | Patient/Lesion |
| PH2 | Dermoscopic | 200 | N/A | Multiclass | N/A | 6.12% | Lesion |

the target datasets into training and validation splits with a ratio of 90%-10%, respectively. Evaluation following a stricter patient-disjoint splitting strategy is discussed in Section F. Both pre-training stages use the AdamW optimizer with a learning rate of $10^{-4}$ and $\lambda = 0.9$. Continual pre-training is performed for 100 epochs on each dataset. For the classification task, we apply linear probing with binary cross entropy loss (BCE) and the AdamW optimization algorithm.

## 4.3 PROTOCOL

The SLIMP model is pre-trained on SLICE-3D, a large-scale medical imaging dataset. For assessing the intrinsic quality of the SLIMP features, evaluation is performed by considering linear probing as well as k-nearest neighbors (kNN) on the downstream skin-lesion classification task on different target datasets (Caron et al., 2021). The skin lesion classification datasets contain different taxonomies, with important class imbalance of varying degrees (Figure 3). To allow consistent comparison across all datasets, we mainly consider the task of classifying lesions as benign and malignant. Performance of the models is evaluated considering four metrics: Accuracy (Acc) and Balanced Accuracy (BA) reported as percentages, and F1-Score and area under the receiver operator curve (AUC) reported as dimensionless numbers. Balanced Accuracy corresponds to the average of the Sensitivity and Specificity scores and is particularly relevant in the medical domain as it captures the model's ability to correctly identify positive and negative instances, even when datasets suffer from significant class imbalance. Section G, provides additional experiments, discussing also the multiclass classification performance of SLIMP and its use in downstream retrieval tasks.

## 4.4 RESULTS

Our main goal is to assess the quality of the skin lesion representations learned by the proposed SLIMP model. Additionally, we examine the extent in which the use of metadata in different parts of the pipeline impacts the performance on the downstream classification task. In these regards, we consider strong baselines in each of these parts. Table 2 presents the results using linear probing on the four target datasets, as well as the macro-averaged metrics further highlighting the generalization ability of the model. The kNN classification results are presented in the Appendix (Table 18).

We first consider comparison using features that have been obtained via pre-training on the reference SLICE-3D dataset. In this context, we consider the Pre-SLIMP setup, which uses the appearance features extracted by the image encoder of SLIMP that is pre-trained on the lesion and patient metadata of SLICE-3D, and compare it against the features obtained by SimCLR (Chen et al., 2020) pre-trained on the images of SLICE-3D. We also consider the downstream classification performance of the subclass-balancing contrastive learning approach (SBCL) proposed in (Hou et al., 2023). We observe that Pre-SLIMP, by exploiting the information encoded in the metadata, achieves similar performance to SimCLR, even though it does not consider any image-based self-supervision. This suggests that SLIMP incorporates information from corresponding metadata in the image representation, leading to more robust representations against image domain shift. By producing more robust features, Pre-SLIMP outperforms SBCL which explicitly handles class imbalance and long-tail distributions.

In addition, Table 2 provides the results from the MAE (He et al., 2022), DINOv2 (Oquab et al., 2023) and BeiTv2 (Peng et al., 2022) generic foundation models, as well as the multi-modal models CLIP (Radford et al., 2021), SigLIP (Zhai et al., 2023), SigLIP-2 (Tschannen et al., 2025) and WhyLesionCLIP (WL-CLIP) (Yang et al., 2024), alongside PanDerm Yan et al. (2025). The latter two serve as robust domain-specific baselines. WL-CLIP is a supervised fine-tuning of CLIP on skin lesion descriptions, while PanDerm is a large-scale vision-language foundation model tailored for clinical dermatology. For a fair comparison, we consider the ViT-B variants of these models, where available. We observe that Pre-SLIMP, via nested image-metadata pre-training achieves a competitive performance against all these models, which have been trained using data that are orders of magnitude larger. Still, the corresponding attention maps (presented in Section I) suggest that SLIMP is better at capturing prominent appearance features of the lesions, hinting that they are more suitable for spatially-aware downstream tasks (e.g. lesion segmentation). Importantly, SLIMP by employing a nested structure dictated by the patient-lesions relation, outperforms massive dermatology-specific foundation models like WL-CLIP and PanDerm (e.g., by more than 6% Balanced Accuracy on average), despite using a significantly smaller backbone. This shows that structured metadata integration helps to increase performance more than simply scaling up vision-language pre-training.

Table 2: Comparison of SLIMP with various baselines, on the lesion classification task using linear probing. MD stands for 'Metadata' used for downstream classification, the asterisk (*) denotes metadata extrapolation from the reference dataset. For all metrics higher values are better. Best results are in **bold**, second best are underlined.

| | MD | PAD-UFES-20 Acc | BA | AUC | HIBA Acc | BA | AUC | HAM10000 Acc | BA | AUC | PH2 Acc | BA | AUC | Average Acc | BA | AUC |
|---|---|---|---|---|---|---|---|---|---|---|---|---|---|---|---|---|
| *Pre-trained Vision Models* | | | | | | | | | | | | | | | | |
| MAE | ✗ | 68.3 | 68.2 | .693 | 79.0 | 79.5 | .848 | 86.0 | 76.7 | .901 | 85.0 | 71.9 | .844 | 79.6 | 74.1 | .822 |
| DINOv2 | ✗ | 76.1 | 77.3 | .828 | 77.2 | 77.4 | .849 | 86.0 | 75.2 | .897 | 86.7 | 81.3 | .867 | 81.5 | 77.5 | .866 |
| BEiTv2 | ✗ | 77.0 | 77.3 | .828 | 79.6 | 79.9 | .851 | 86.2 | 78.2 | .906 | 95.0 | 96.4 | .992 | 84.5 | 83.1 | .894 |
| PanDerm$_{Base}$ | ✗ | 75.2 | 75.5 | .857 | 86.4 | 86.5 | .932 | 85.1 | 70.0 | .892 | 95.0 | 96.4 | .969 | 85.4 | 82,1 | .913 |
| PanDerm$_{Large}$ | ✗ | 80.0 | 80.1 | .904 | 87.7 | 87.7 | .941 | 88.5 | 77.6 | .925 | 95.0 | 96.4 | **1.00** | 87.8 | 85.5 | .943 |
| *Pre-trained Vision-Language Models* | | | | | | | | | | | | | | | | |
| CLIP | ✗ | 70.9 | 71.0 | .795 | 82.1 | 82.5 | .893 | 85.4 | 78.7 | .892 | 90.0 | 84.4 | .891 | 82.1 | 79.2 | .868 |
| SigLIP | ✗ | 74.8 | 75.0 | .823 | 76.5 | 76.9 | .838 | 86.5 | 79.0 | .900 | 95.0 | 96.4 | .969 | 83.2 | 81.9 | .883 |
| SigLIP-2 | ✗ | 77.8 | 77.9 | .853 | 82.1 | 82.6 | .856 | 86.8 | 79.8 | .907 | 95.0 | 96.4 | **1.00** | 85.4 | 84.3 | .904 |
| WL-CLIP | ✗ | 81.7 | 81.9 | .883 | 82.1 | 82.2 | .896 | **88.7** | 83.1 | **.929** | 90.0 | 93.8 | **1.00** | 85.6 | 85.2 | .927 |
| *Pre-trained on SLICE-3D* | | | | | | | | | | | | | | | | |
| SimCLR | ✗ | 70.4 | 70.5 | .766 | 84.6 | 84.3 | .913 | 81.2 | 69.4 | .868 | 95.0 | 87.5 | **1.00** | 82.8 | 77.9 | .849 |
| SBCL | ✗ | 66.1 | 66.0 | .672 | 66.7 | 67.5 | .671 | 56.0 | 63.8 | .710 | 75.0 | 75.0 | .734 | 66.0 | 68.1 | .684 |
| Pre-SLIMP | ✗ | 76.5 | 76.0 | .781 | 75.9 | 76.0 | .845 | 83.6 | 67.8 | .855 | 90.0 | 83.3 | .941 | 81.5 | 75.8 | .827 |
| Ret-SLIMP | ✓* | 77.0 | 77.0 | .814 | 81.5 | 81.3 | .861 | 82.2 | 71.0 | .836 | 95.0 | 93.8 | .969 | 83.9 | 80.8 | .837 |
| *Continual pre-training* | | | | | | | | | | | | | | | | |
| SBCL-C | ✗ | 71.3 | 71.1 | .711 | 72.2 | 73.9 | .760 | 62.2 | 73.4 | .816 | 90.0 | 84.4 | .719 | 73.9 | 75.7 | .762 |
| SLIMP$_{IMAGE}$ | ✗ | 76.1 | 75.5 | .807 | 77.8 | 78.1 | .867 | 84.7 | 69.2 | .889 | 95.0 | 96.4 | .988 | 83.4 | 79.8 | .854 |
| SLIMP$_{FLAT}$ | ✓ | 85.7 | 85.3 | .906 | 84.6 | 84.5 | .911 | 84.4 | 75.6 | .894 | **100** | **100** | **1.00** | 87.4 | 85.5 | .904 |
| SLIMP$_{B=4}$ | ✓ | 90.9 | 90.2 | .926 | 92.0 | 91.9 | .954 | 87.3 | 83.5 | .923 | **100** | **100** | **1.00** | 92.6 | 91.4 | **.951** |
| SLIMP$_{B=8}$ | ✓ | 90.9 | 90.5 | .929 | **92.6** | **92.4** | .944 | 87.7 | **84.5** | .929 | **100** | **100** | **1.00** | 92.8 | 91.9 | **.951** |
| *Supervised* | | | | | | | | | | | | | | | | |
| TFormer | ✓ | **91.3** | **91.3** | **.960** | 88.9 | 88.9 | **.963** | 82.1 | 76.2 | .875 | 95.0 | 91.7 | .988 | 89.3 | 87.0 | .947 |
| *Low-shot Evaluation* | | | | | | | | | | | | | | | | |
| SLIMP$_{1\%}$ | ✓ | 83.9 | 84.1 | .908 | 75.3 | 75.8 | .863 | 78.7 | 73.8 | .847 | 70.0 | 64.3 | .548 | 77.0 | 74.5 | .792 |
| SLIMP$_{10\%}$ | ✓ | 88.7 | 88.2 | .922 | 84.0 | 84.2 | .917 | 83.9 | 77.8 | .887 | 90.0 | 84.5 | .952 | 86.6 | 83.7 | .920 |
| TFormer$_{1\%}$ | ✓ | 81.3 | 81.2 | .880 | 74.7 | 74.7 | .811 | 81.9 | 66.0 | .804 | 35.0 | 48.8 | .702 | 68.2 | 67.7 | .799 |
| TFormer$_{10\%}$ | ✓ | 85.2 | 85.1 | .886 | 82.1 | 81.7 | .876 | 81.5 | 65.1 | .858 | 90.0 | 83.3 | .810 | 84.7 | 78.8 | .857 |

**Use of metadata** As the metadata attributes of the target datasets differ from the reference one, the pre-trained metadata encoders cannot be directly used. This shortcoming is addressed by the SLIMP model, which applies continual pre-training on the target dataset as described in Section 4.2. This allows the use of target dataset metadata, both at the continual pretraining stage and at the downstream classification task. We see that the image representations obtained after continual pre-training, denoted as SLIMP$_{IMAGE}$, offer improved performance compared to Pre-SLIMP, clearly outperforming the SBCL method continually pre-trained on the target datasets (SBCL-C). Importantly, the complete SLIMP method, which uses the features obtained by all data modalities in the downstream task, leads to significantly improved performance on average and across most of the datasets. Increasing the patient batch size from 4 to 8 offers some marginal improvement. Interestingly, SLIMP also shows competitive performance compared to TFormer (Zhang et al., 2023), a fully supervised model for multi-modal lesion classification trained directly on both the images and metadata of the target dataset, showing a decrease in performance only for the PAD-UFES-20 dataset.

The use of pseudo-modalities constructed through retrieval of metadata from the reference dataset, denoted as Ret-SLIMP in the tables, shows consistently improved performance compared to Pre-SLIMP and comparable performance with SLIMP$_{IMAGE}$, even though it has not seen any data from the target datasets during training. This is valuable when the target dataset lacks metadata. This observation also further highlights the importance of using metadata for downstream classification.

**Impact of nested contrastive learning** To assess the effectiveness of the nested contrastive learning employed by SLIMP, we also consider a variant of SLIMP, SLIMP$_{FLAT}$, which comprises a single InfoNCE loss applied between the image features and the features obtained by a tabular encoder operating on the concatenated patient-lesion metadata. SLIMP clearly outperforms this single-level variant, demonstrating the effectiveness of its nested contrastive learning architecture in capturing image-metadata relations. The only exception is PH2, where both variants converge as the dataset does not contain patient-level metadata.

Table 3 offers a more detailed analysis by examining the two variants of the SLIMP architecture (FLAT and NESTED) when trained on each dataset from scratch. The results clearly show that the variant based on nested contrastive learning achieves significantly higher performance compared to the one that uses the same metadata using a single contrastive learning stage. This is attributed to the implicit grouping of each patient's lesions, producing features that better capture their phenotype. The same table reports the difference of each metric regarding the SLIMP model, showing that the pre-training on the SLICE-3D dataset helps to achieve improved performance across all datasets.

**Low-shot evaluation** The proposed multi-modal continual pre-training strategy does not rely on target labels. This is crucial as data labeling is expensive and time-consuming, especially in the context of skin lesion classification and other similar medical applications. To further assess the quality of the learned representations, we examine how SLIMP performs in a low-shot learning setting, considering that only 1% or 10% of the target dataset labels are available for downstream classification. The results, presented in the last rows of Table 2 (highlighted in orange), indicate that the SLIMP features lead to remarkable low-shot learning performance. It is interesting to note that in most cases, SLIMP low-shot performance is better than $SLIMP_{IMAGE}$ and $SLIMP_{FLAT}$. The first suggests the importance of the model making use of metadata both during pre-training and also for the downstream classification task. Comparable performance to $SLIMP_{FLAT}$ further highlights the ability of the nested contrastive learning to capture relations among metadata and images.

### 4.5 ABLATION

To assess the importance of incorporating two distinct levels of metadata, we compare different variants of SLIMP in Table 4. Specifically, in the first row we consider the linear probing performance of a variant where only the output features of the image encoder are utilized for downstream classification on the target dataset. In the second row we consider both the features of the image encoder and the lesion-level tabular metadata encoder. The third row shows the results of the proposed SLIMP model. The last three rows report analogous results with kNN classification. The results suggest that the addition of each modality contributes positively to the downstream task performance. Additional ablations are provided in Section F.

## 5 CONCLUSIONS AND LIMITATIONS

We have presented SLIMP, a novel nested multi-modal pre-training strategy for learning rich skin lesion representations by considering lesion images in combination with associated lesion-level as well as patient-level metadata. The experimental evaluation demonstrates SLIMP's ability to learn representations that improve performance in downstream classification tasks, by combining information about the patient's lesion phenotype, with information regarding their traits and habits. In this context, we propose strategies for fully exploiting available metadata, through all the stages of the learning process, including a method that enables the enhancement of image-only skin lesion datasets by 'borrowing' patient and lesion metadata from reference pre-training data. Importantly, the proposed method does not rely on data annotations, handling a major challenge in healthcare applications where data annotation incurs significant costs. The results obtained for low-shot settings of the target datasets, demonstrate the quality of the obtained skin lesion representations as they enable high classification performance even with minimal labeled data. Considering the above, our proposed method has the potential to become widely applicable in clinical settings, providing insights and decision support during skin lesion diagnosis.

Despite its strengths, the proposed method has certain limitations. Firstly, the nested pre-training strategy requires a data structure that incorporates both patient- and lesion-level metadata, which

Table 3: Comparison of flat (single-level contrastive loss) and nested SLIMP architecture when trained on each dataset separately. The difference with $SLIMP_{B=4}$ model is reported in superscript.

| | PAD-UFES-20 | | | HIBA | | | HAM10000 | | |
|---|---|---|---|---|---|---|---|---|---|
| Architecture | Acc | BA | AUC | Acc | BA | AUC | Acc | BA | AUC |
| FLAT | $85.2^{(-5.7)}$ | $84.7^{(-5.5)}$ | $.902^{(-.024)}$ | $85.2^{(-6.8)}$ | $84.9^{(-7.0)}$ | $.915^{(-.039)}$ | $77.8^{(-9.5)}$ | $78.6^{(-4.9)}$ | $.860^{(-.063)}$ |
| NESTED | $87.4^{(-3.5)}$ | $86.9^{(-3.3)}$ | $.910^{(-.016)}$ | $88.9^{(-3.1)}$ | $88.7^{(-3.2)}$ | $.934^{(-.020)}$ | $83.6^{(-3.7)}$ | $82.8^{(-0.7)}$ | $.901^{(-.022)}$ |

Table 4: Ablation study of the SLIMP encoder outputs used for downstream classification.

| Image | Metadata | | PAD-UFES-20 | | | | HIBA | | | | HAM10000 | | | | PH2 | | | |
|---|---|---|---|---|---|---|---|---|---|---|---|---|---|---|---|---|---|---|
| | Lesion | Patient | Acc | BA | F1 | AUC | Acc | BA | F1 | AUC | Acc | BA | F1 | AUC | Acc | BA | F1 | AUC |
| *Linear Probing* | | | | | | | | | | | | | | | | | | |
| ✓ | ✗ | ✗ | 76.1 | 75.5 | 0.764 | 0.807 | 77.8 | 78.1 | .763 | .867 | 84.7 | 69.2 | .529 | .889 | 95.0 | 96.4 | .923 | .988 |
| ✓ | ✓ | ✗ | 89.6 | 89.1 | .908 | .908 | 88.3 | 88.1 | .891 | .947 | 85.3 | 74.1 | .599 | .899 | **100** | **100** | **1.00** | **1.00** |
| ✓ | ✓ | ✓ | **90.9** | **90.2** | **.921** | **.926** | **92.0** | **91.9** | **.925** | **.954** | **87.3** | **83.5** | **.707** | **.923** | - | - | - | - |
| *kNN* | | | | | | | | | | | | | | | | | | |
| ✓ | ✗ | ✗ | 74.8 | 74.6 | .770 | .633 | 82.7 | 82.6 | .835 | .770 | 84.1 | 69.4 | .528 | .851 | **95.0** | 91.7 | .909 | **1.00** |
| ✓ | ✓ | ✗ | 90.0 | 89.7 | .911 | .865 | 87.0 | 86.7 | .884 | .812 | **85.8** | 76.1 | .626 | .886 | 95.0 | 96.4 | .923 | .940 |
| ✓ | ✓ | ✓ | **93.5** | **93.2** | **.942** | **.911** | 87.7 | 87.5 | .885 | .849 | 85.8 | 78.0 | .645 | .891 | - | - | - | - |

may limit its adaptability to other domains where such structured scenarios do not straight-forwardly exist. Secondly, significant shift in the image domain, including high variability in the sources and resolutions of lesion images, can possibly downgrade downstream performance. This problem can be addressed by incorporating image augmentations in the learning process. Regarding negative impacts, it should be noted that misuse of this method, as for all computer-aided diagnosis methods, can lead to overdiagnoses, or misdiagnoses, with important psychological and economic repercussions. Hence, real-life use of such systems should be intended only for assisting the decision-making of expert users, and not for direct use by the patients.

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

## A NOTATION

Table 5 summarizes the notation used throughout the manuscript.

Table 5: Summary of the notation.

| Notation | Description |
|---|---|
| $M$ | Number of patients indexed by $p \in \{1, ...M\}$ |
| $N_p$ | Total lesions of patient $p$ indexed by $l \in \{1, ...N_p\}$ |
| $P_p$ | Tabular metadata for patient $p$ |
| $L_p^l$ | Tabular metadata for lesion $l$ of patient $p$ |
| $I_p^l$ | Lesion image $l$ of patient $p$ |
| $w_p^l$ | Image encoder output of $I_p^l$ |
| $h_p^l$ | Tabular encoder output of $L_p^l$ |
| $x_p$ | Tabular encoder output of $P_p$ |
| $z_p$ | Linearly transformed output based on $\{w_p^l, h_p^l\}$ |
| $D$ | Dimensionality of each embedding |
| $\tilde{H} = \{\tilde{h}^l\}_{l=1}^N$ | Lesion-level pre-trained features of original dataset |
| $\tilde{X} = \{\tilde{x}_p\}_{p=1}^M$ | Patient-level pre-trained features of original dataset |
| $\hat{h}^{l'}$ | Retrieved features from $\tilde{H}$ |
| $\hat{z}_p$ | Linearly transformed output based on $\{w_p^l, \hat{h}^{l'}\}$ |
| $\hat{x}_{p'}$ | Retrieved features from $\tilde{X}$ |
| $\hat{y}_p^l$ | concat$\{w_p^l, \hat{h}^{l'}, \hat{x}_{p'}\}$ |

## B DATASET DETAILS

The following skin-lesion classification datasets are considered:

**SLICE-3D** (Kurtansky et al., 2024): a public skin lesion dataset containing up to 401,059 15mm-by-15mm field-of-view cropped images, centered on distinct lesions extracted from 3D Total Body Photography (TBP) collected across seven dermatologic centers worldwide. The dataset was curated for the ISIC 2024 Challenge and contains 40 clinical features concerning both patients and lesions, such as age, sex, general anatomic site, common patient identifier, clinical size, and various data fields from the TBP Lesion Visualizer.

**PAD-UFES-20** (Pacheco et al., 2020): a skin lesion dataset containing 2,298 close-up clinical images collected using different smartphone devices. It includes six types of skin lesions, data from 1,373 patients, and up to 22 clinical features per sample, covering both patient and lesion attributes, such as age, skin lesion location, and lesion diameter. The skin lesions are: Basal Cell Carcinoma (BCC), Squamous Cell Carcinoma (SCC), Actinic Keratosis (ACK), Seborrheic Keratosis (SEK), Melanoma (MEL), and Nevus (NEV).

**HIBA** (ISIC, 2024): a skin lesion archive with clinical and dermoscopic images collected in Argentina, containing 1,616 images of 10 different types of skin lesions, including Basal Cell Carcinoma (BCC), Squamous Cell Carcinoma (SCC), Actinic Keratosis (ACK), Seborrheic Keratosis (SEK), Melanoma (MEL), Nevus (NEV), Vascular Lesion (VASC), Lichenoid Keratosis (LK), Solar Lentigo (SL), and Dermatofibroma (DF).

**HAM10000** (Tschandl et al., 2018): also known as "Human Against Machine with 10,000 training images," this dataset comprises 10,015 multi-source dermoscopic images of skin lesions divided into seven classes and includes four clinical features, with two related to patient demographics and two describing lesion characteristics. The skin lesions are: Actinic Keratosis and Intraepithelial Carcinoma (AKIEC), Basal Cell Carcinoma (BCC), Benign Keratosis-like Lesions (BKL), Dermatofibroma (DF), Melanoma (MEL), Melanocytic Nevi (NV), and Vascular Lesions (VASC).

**PH$^2$** (Mendonça et al., 2013): a small dataset with 200 dermoscopic skin lesion images, including three classes: 80 common nevi, 80 atypical nevi, and 40 melanomas. The dataset contains 13 clinical

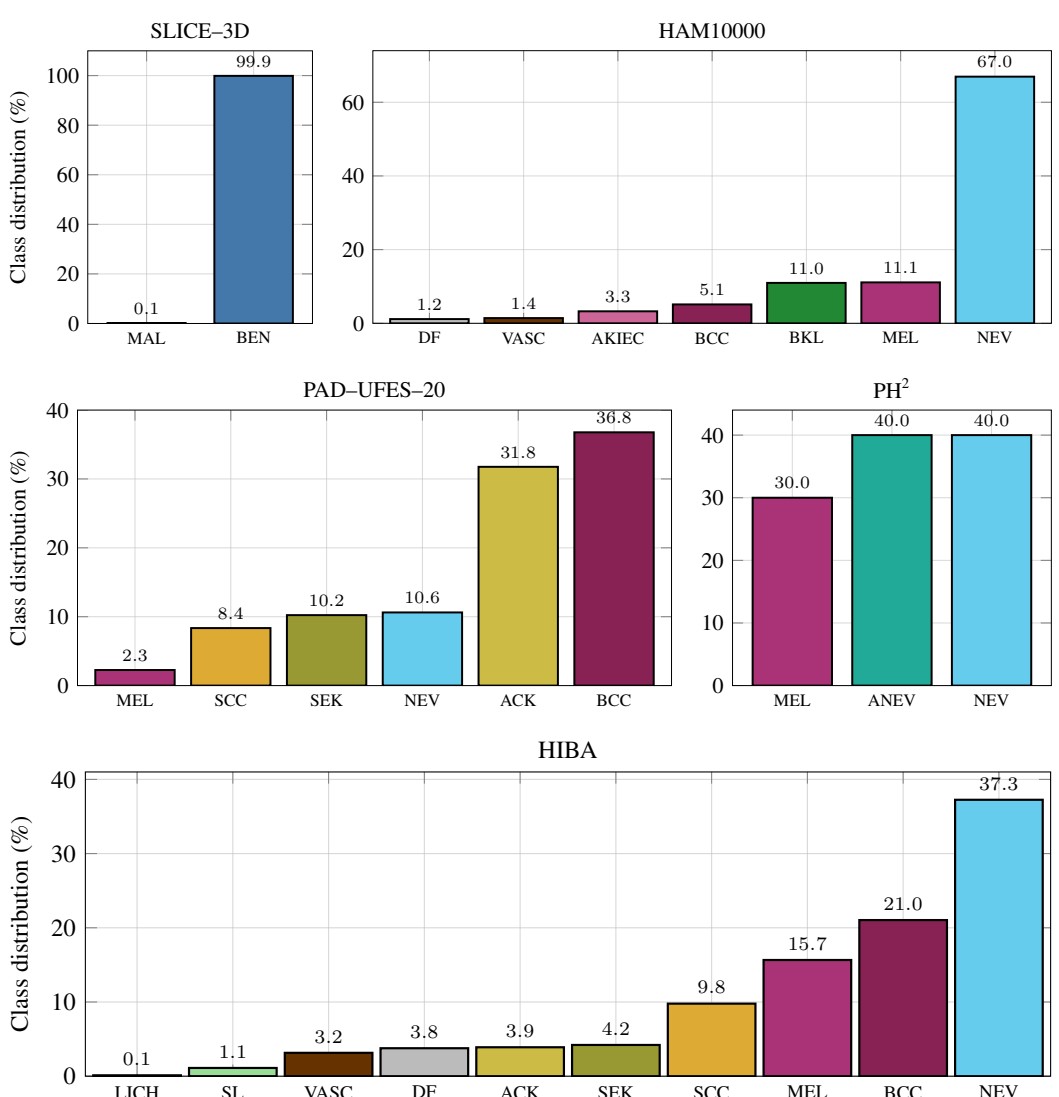

Figure 3: Class distribution within each dataset considered.

lesion features, such as clinical and histological diagnosis, and the assessment of various dermoscopic criteria.

**SLICE-3D** (Kurtansky et al., 2024), being the largest and most complete one, is considered as the reference dataset for pre-training the SLIMP model. All other datasets are considered as target datasets for performing skin classification using the pretrained model. Unless otherwise stated, evaluation is performed considering binary classification targets (benign/malignant) of the datasets that are better balanced.

For **PAD-UFES-20** (Pacheco et al., 2020), malignant classes include Basal Cell Carcinoma (BCC), Melanoma (MEL) and Squamous Cell Carcinoma (SCC), while benign classes include Actinic Keratosis (ACK), Nevus (NEV) and Seborrheic Keratosis (SEK). In **HAM10000** (Tschandl et al., 2018), Basal Cell Carcinoma (BCC) and Melanoma (MEL) are categorized as malignant, with benign classes comprising Actinic Keratosis (ACK), Nevus (NEV), Vascular Lesion (VASC), Dermatofibroma (DF), and Benign Keratosis-like Lesions (BKL). In **HIBA** (ISIC, 2024), the malignant class includes Basal Cell Carcinoma (BCC), Melanoma (MEL) and Squamous Cell Carcinoma (SCC), while benign lesions encompass Actinic Keratosis (ACK), Dermatofibroma (DF), Lichenoid Keratosis (LK), Seborrheic Keratosis (SEK), Nevus (NEV), Vascular Lesion (VASC), and Solar Lentigo (SL). In the case

of **PH2** (Mendonça et al., 2013) dataset, the malignant category consists only of melanomas, while common nevi and atypical nevi were grouped as benign. **SLICE-3D** (Kurtansky et al., 2024), the largest dataset in this study, is inherently binary, with an extremely imbalanced distribution: $99.9\%$ of lesions are benign, while only $0.1\%$ are malignant.

## C  NESTED CONTRASTIVE LOSS

Letting $s(\cdot, \cdot)$ denote the cosine similarity function and $\tau$ a temperature parameter, the two-level nested contrastive loss with a weighting factor $\lambda \in [0, 1]$ is defined as follows:

$$\mathcal{L}_{lesions}^{p} = -\frac{1}{2N_p} \sum_{l=1}^{N_p} \left( \log \frac{\exp(\mathrm{s}(w_p^l, h_p^l)/\tau)}{\sum_{j \in N_p} \exp(\mathrm{s}(w_p^l, h_p^j)/\tau)} + \log \frac{\exp(\mathrm{s}(h_p^l, w_p^l)/\tau)}{\sum_{j \in N_p} \exp(\mathrm{s}(h_p^j, w_p^l)/\tau)} \right), \quad (1)$$

$$\mathcal{L}_{patient} = -\frac{1}{2M} \sum_{p=1}^{M} \left( \log \frac{\exp(\mathrm{s}(z_p, x_p)/\tau)}{\sum_{i \in M} \exp(\mathrm{s}(z_p, x_i)/\tau)} + \log \frac{\exp(\mathrm{s}(x_p, z_p)/\tau)}{\sum_{i \in M} \exp(\mathrm{s}(x_i, z_p)/\tau)} \right), \quad (2)$$

$$\mathcal{L}_{total} = \frac{\lambda}{M} \sum_{p=1}^{M} \mathcal{L}_{lesions}^{p} + (1 - \lambda)\mathcal{L}_{patient}. \quad (3)$$

$\mathcal{L}_{lesions}$ and $\mathcal{L}_{patient}$ treat features from the same lesion or patient, respectively, as positive pairs while pushing apart features originating from different lesions or patients.

## D  ADDITIONAL TRAINING DETAILS

**Batch sampling strategy**   For both the initial and continual self-supervised pre-training stages, we construct each batch with $B$ patients, including their respective patient-level tabular metadata. Additionally, for each patient, we sample $N$ lesion images and their corresponding lesion-level tabular metadata. The number of lesions $N$ varies per patient and is capped by an upper limit $N_{max}$. If a patient has more lesions, then a subset of $N = N_{max}$ lesions is randomly sampled in each epoch. In addition, a positive lesion sampling strategy is implemented, ensuring that, if a patient has malignant lesions, they are always included in the $N$ lesions sampled during training. This ensures that the model encounters an adequate number of malignant lesions. Section F provides a relative ablation.

For the retrieval-based extrapolation setup, where the images from the target dataset lack both lesion and patient metadata, we create two independent pools with tabular features derived from the metadata of the SLICE-3D reference dataset by passing them through the pre-trained inner and outer tabular encoders. This step does not preserve any association between patients and their corresponding lesions. Consequently, the retrieval process of patient/lesion-level metadata is not constrained to select features from the same patient across every modality, maximizing the flexibility of the proposed architecture.

**Training details of SLIMP**   During self-supervised training, both in the initial pre-training on SLICE-3D and in the continual pre-training on the target dataset, SLIMP uses the same nested lesion-level and patient-level InfoNCE objectives ($\mathcal{L}_{lesions}$ and $\mathcal{L}_{patient}$). During reference pre-training on the SLICE-3D dataset, all components of the architecture, including the ViT backbone and the lesion- and patient-metadata encoders, are fully optimized. Instead, during self-supervised continual pre-training, SLIMP is adapted on the unlabeled data of the target datasets by fine-tuning the embedding layers of the image and metadata encoders while keeping all other layers frozen. Specifically, the ViT patch-embedding layer and the input-embedding layers of the two TRACE tabulat encoders are fine-tuned, keeping the rest of the blocks frozen. This strategy enables a slight but effective domain-adaptation as shown in Table 14 where it consistently outperforms fine-tuning all the model parameters. Both pre-training stages are entirely class-agnostic.

For supervised downstream skin-lesion classification, all encoders remain frozen and a linear classifier is trained, as defined in a standard Linear Probing schema, with a *Cross-Entropy* loss. If the target dataset misses patient-level and/or lesion-level metadata, we first apply retrieval-based metadata

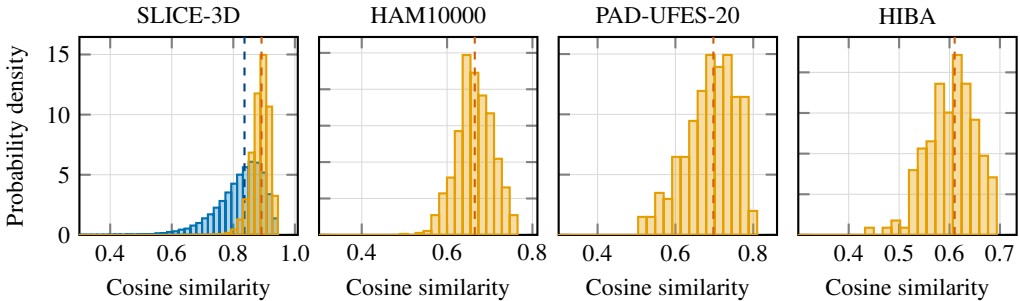

Figure 4: Cosine similarity distributions between image and metadata representations. On SLICE-3D validation hold-out set, we report the similarity to the ground-truth metadata and to metadata retrieved from the training set using SLIMP. For HAM10000, PAD-UFES-20, and HIBA, the retrieved-metadata distributions are shown. Dashed vertical lines indicate the median similarity for each distribution.

extrapolation at inference time without updating any encoder parameters (see Section 3), and then apply linear probing using the resulting multi-modal features.

**Training details of supervised methods**   We pre-train SBCL (Hou et al., 2023) with a ResNet-32 architecture, for 1000 epochs on the SLICE-3D dataset, followed by a dataset-specific continual pre-training (SBCL-C) for 100 epochs. Both pre-training setups use the SGD optimizer with a learning rate of $0.5$ for the initial pre-training and $1e^{-2}$ for the continual pre-training. We evaluate each target dataset on the corresponding SBCL-C model by applying linear classification for 150 epochs (following the SLIMP linear probing setting) with a learning rate of $0.1$. During linear classification, we select the Classifier-Balancing (CB) (Kang et al., 2020) train rule, which proved to outperform LDAM (Label-Distribution-Aware Margin Loss) (Cao et al., 2019).

Regarding TFormer (Zhang et al., 2023), we utilize the variant designed to process two modalities, namely clinical images and tabular metadata, since the target datasets do not explicitly provide clinical and dermoscopic image pairs of the same lesion. During training, TFormer was fine-tuned on each target dataset, using the Adam optimizer with a learning rate of $1e^{-4}$, and a weight decay of $1e^{-4}$. The learning rate was adjusted dynamically through the Cosine Annealing learning rate scheduler. The loss function used throughout the training process was Binary Cross-Entropy.

# E  STRUCTURE OF EMBEDDING SPACE

Table 6 reports the distribution percentiles of the cosine similarity between the image features with the matching (positive) and non-matching (negative) metadata embeddings on

Table 6: Percentiles of cosine similarity between image features and matching vs. non-matching metadata embeddings on SLICE-3D.

| Percentile | 2% | 10% | 25% | 50% | 75% | 90% | 98% |
|---|---|---|---|---|---|---|---|
| Non-Matching | -0.328 | -0.213 | -0.117 | -0.006 | 0.109 | .217 | 0.369 |
| Matching | 0.614 | 0.717 | 0.780 | 0.836 | 0.878 | 0.906 | 0.931 |

the SLICE-3D dataset, noting that each of them is well approximated by a unimodal, almost symmetric distribution. Importantly, the distribution of the negative pairs lies far away from the distribution of the positive pairs, showing a significant separability in the embedding space, indicating that a well-structured representation space has been recovered during the pre-training phase.

To provide some further insight, we consider a small subset of SLICE-3D (10%) as a validation set and we produce the distribution of the similarity scores between the images of this set with the matching metadata in the embedding space, as well as the corresponding distribution of the similarity scores with the most similar metadata retrieved from the training set. The distributions are shown in Figure 4, suggesting that there is a good agreement between them. Moreover, Figure 4 presents the similarity score distributions between the images from the targets datasets and the retrieving metadata from the SLICE-3D reference dataset. Although these distributions, as expected, are shifted towards lower scores, still the alignment between the image-metadata representations is quite satisfactory.

This in part explains why the proposed metadata extrapolation method can lead to improved results, as can be seen by the comparison between Pre-SLIMP and Ret-SLIMP in Table 2.

Moreover, Table 7 reports the Recall@k metrics on the SLICE-3D validation set to directly assess whether the true metadata associated with a given image is among the top retrieved candidates. The fact that R@1

Table 7: Image-metadata retrieval results on SLICE-3D validation set. R@k: Recall at rank k.

| R@1 | R@5 | R@10 | R@15 | R@20 | R@100 |
|---|---|---|---|---|---|
| 45.1 | 76.6 | 86.3 | 90.5 | 92.8 | 99.1 |

exceeds 45% demonstrates that the model retrieves the correct metadata as the top match nearly half of the time. Given that the validation set contains over 40,000 samples, this indicates that the model is capturing important alignment cues between image and metadata modalities. The rapid increase between R@1 and R@5/R@10 further indicates that the matching metadata is usually found within a very narrow ranking window, reflecting a well-structured embedding space. Notably, R@100 reaches 99%, an important result given the size of the validation set.

## F EXTENDED ABLATION

We report additional ablations concerning the choice of image and tabular encoders, as well as the patient batch size. In the tables below, we highlight in light blue the reference configuration adopted in the experiments of the main text.

### F.1 IMAGE ENCODER

We consider the influence of the image encoder size on the downstream skin lesion classification task. Specifically, we consider the Tiny, Small & Base ViT variants (Dosovitskiy et al., 2021; Touvron et al., 2021). Table 8 shows the influence of the image encoder size on the performance metrics across four datasets: PAD-UFES-20, HIBA, HAM10000, and PH2. Interestingly, the influence of the image encoder size in the case of SLIMP is reduced, which can be attributed to the complementary information added by the metadata through the tabular encoder. Table 9 reports the number of parameters for the different image encoder sizes, with ViT-Base being approximately $4\times$ larger than ViT-Small and $15\times$ larger than ViT-Tiny.

Table 8: Impact of image encoder size on the skin classification performance using SLIMP. Best results in **bold**.

| | | PAD-UFES-20 | | | | HIBA | | | | HAM10000 | | | | PH2 | | | |
|---|---|---|---|---|---|---|---|---|---|---|---|---|---|---|---|---|---|
| | | Acc | BA | F1 | AUC | Acc | BA | F1 | AUC | Acc | BA | F1 | AUC | Acc | BA | F1 | AUC |
| linprob | SLIMP w/ ViT-T | 89.6 | 89.0 | .908 | .922 | 89.5 | 89.3 | .904 | .939 | 84.7 | 81.7 | .665 | .910 | 95.0 | 91.7 | .909 | **1.00** |
| | SLIMP w/ ViT-S | **90.9** | **90.2** | **.921** | **.926** | **92.0** | **91.9** | **.925** | **.954** | **87.3** | **83.5** | **.707** | **.923** | **100** | **100** | **1.00** | **1.00** |
| | SLIMP w/ ViT-B | 87.8 | 86.9 | .896 | .899 | 83.3 | 83.0 | .851 | .918 | 81.7 | 72.4 | .553 | .862 | 90.0 | 83.3 | .800 | **1.00** |
| kNN | SLIMP w/ ViT-T | 81.7 | 81.4 | .837 | .858 | 83.3 | 83.2 | .842 | .904 | 85.7 | 74.9 | .612 | **.904** | 90.0 | 83.3 | .800 | **1.00** |
| | SLIMP w/ ViT-S | **93.5** | **93.2** | **.942** | **.911** | **87.7** | **87.5** | **.885** | **.849** | **85.8** | **78.0** | **.645** | .893 | 95.0 | **96.4** | **.923** | .940 |
| | SLIMP w/ ViT-B | 84.8 | 84.4 | .865 | .900 | 81.5 | 81.3 | .830 | .887 | 82.3 | 64.4 | .438 | .851 | 80.0 | 66.7 | .500 | **1.00** |

Table 9: Number of parameters (millions) for the proposed SLIMP method for different image and tabular encoders.

| | | w/ TRACE | | w/ FT-Transformer |
|---|---|---|---|---|
| | ViT-Tiny | ViT-Small | ViT-Base | ViT-Small |
| SLICE-3D | 8.7 | 34.3 | 136 | 99.9 |
| PAD-UFES-20 | 2.2 | 8.3 | 32.6 | |
| HIBA | 2.1 | 8.0 | 31.3 | 78.5 |
| HAM10000 | 2.1 | 8.0 | 31.3 | |

The choice of $N$, the number of images and lesions selected per patient during training, also plays a role in performance differences. For ViT-Tiny and ViT-Small, $N = 100$ was chosen to balance computation and training efficiency, while for ViT-Base, $N = 50$ was used due to the model's significantly larger size and computational requirements. This may partially explain the performance drop observed in ViT-Base architectures, as the model has less diverse per-patient data for training. In summary, ViT-Small tends to strike the best balance between performance and model complexity, as seen across most datasets.

Table 10: Comparison between the generic tabular encoder FT-Transformer and the tabular encoder for medical data TRACE. Best results in **bold**.

| | | PAD-UFES-20 | | | | HIBA | | | | HAM10000 | | | |
|---|---|---|---|---|---|---|---|---|---|---|---|---|---|
| | | Acc | BA | F1 | AUC | Acc | BA | F1 | AUC | Acc | BA | F1 | AUC |
| kNN linprob | SLIMP w/ FT-Transformer | 89.6 | 89.1 | .908 | **.946** | 84.6 | 84.0 | .871 | .910 | 80.2 | 50.0 | .000 | .655 |
| | SLIMP w/ TRACE | **90.9** | **90.2** | **.921** | .926 | **92.0** | **91.9** | **.925** | **.954** | **87.3** | **83.5** | **.707** | **.923** |
| | SLIMP w/ FT-Transformer | 87.4 | 87.2 | .886 | **.939** | 82.7 | 82.6 | .837 | .882 | 77.7 | 52.4 | .159 | .745 |
| | SLIMP w/ TRACE | **93.5** | **93.2** | **.942** | .911 | **87.7** | **87.5** | **.885** | **.849** | **85.8** | **78.0** | **.645** | **.893** |

## F.2 TABULAR ENCODER

We compare the performance of SLIMP considering two tabular encoders: FT-Transformer (Gorishniy et al., 2021) and TRACE (Christopoulos et al., 2025). Table 10 presents the corresponding performance across all datasets, using ViT-Small as the image encoder. TRACE, which is specialized for clinical data, consistently outperforms the generic FT-Transformer across all datasets and metrics considered, despite the fact that SLIMP with FT-Transformer has a significantly larger number of parameters, as shown in Table 9. In fact, despite being over four times bigger, FT-Transformer does not achieve the same level of performance. Moreover, in contrast to the adopted tabular encoder TRACE, FT-Transformer requires a significant amount of hyperparameter tuning to achieve optimal performance. These observations suggest that the task-specific design of TRACE offers a better balance of efficiency and performance when working with medical metadata, making it a more suitable choice for SLIMP.

Table 11 compares the computational complexity, measured in GFLOPS, for SimCLR, SLIMP with FT-Transformer, and SLIMP with TRACE with different encoder sizes (ViT-Tiny, ViT-Small, ViT-Base). Naturally, computational costs scale with the size of the ViT encoder, highlighting the trade-off between model size and ef-

Table 11: Comparison of computational complexity in terms of GFLOPS between SBCL(-C), TFormer, SimCLR, SLIMP with FT-Transformer, and SLIMP with TRACE with different encoder sizes. ViT-T, ViT-S and ViT-B correspond to ViT-Tiny, ViT-Small and ViT-Base, respectively.

| | GFLOPS |
|---|---|
| SBCL(-C) | 0.564 |
| TFormer | 4.509 |
| SimCLR | 1.258 \| 4.608 \| 17.582 (ViT-T \| ViT-S \| ViT-B) |
| SLIMP w/ FT-Transformer | 1.694 \| 6.298 \| 24.233 (ViT-T \| ViT-S \| ViT-B) |
| SLIMP w/ TRACE | 1.298 \| 4.765 \| 18.203 (ViT-T \| ViT-S \| ViT-B) |

ficiency. In relation to metadata encoding, SimCLR which lacks metadata encoding, is slightly more efficient compared to the proposed multimodal SLIMP method, but SLIMP generally performs better, as has been shown in the results presented in the main text. On the other hand, the FT-Transformer tabular encoder introduces a significant overhead. The reference configuration featuring SLIMP with TRACE is a more balanced choice, offering improved performance with significantly less GFLOPS compared to the FT-Transformer. The number of GFLOPS for the supervised approaches SBCL, SBCL-C and TFormer are also reported in the table for comparison. Additionally, Table 12, reports the number of parameters and the relative training time between SimCLR, SLIMP, SBCL and TFormer. Relative training times are normalized with respect to the SimCLR's training time on SLICE-3D.

Table 12: Model size comparison based on the total trainable parameters for every dataset (columns) and the relative training time, normalized to SimCLR's training time on SLICE-3D.

| | | SLICE-3D | PAD-UFES-20 | HIBA | HAM10000 | PH2 |
|---|---|---|---|---|---|---|
| # params | SimCLR | 5.5M | | | | |
| | SLIMP | 34.3M | 8.3M | 8.0M | 8.0M | 4.1M |
| | SBCL | 0.5M | 0.5M | 0.5M | 0.5M | 0.5M |
| | TFormer | | 27.8M | 27.8M | 27.8M | 27.8M |
| rel. time | SimCLR | 1 | | | | |
| | SLIMP | 0.3 | 0.04 | 0.03 | 0.1 | 0.002 |
| | SBCL | 0.2 | 0.06 | 0.05 | 0.01 | 0.002 |
| | TFormer | | 0.01 | 0.01 | 0.04 | 0.002 |

## F.3 Positive sampling strategy

To assess the impact of the positive sampling strategy, we train a SLIMP variant on SLICE-3D with uniform lesion sampling (w/o p.s). As reported in Table 13, positive sampling (w/ p.s) provides small, yet consistent perfor-

Table 13: Uniform (w/o p.s) vs. positive sampling (w/ p.s) ablation on PAD-UFES-20, HIBA and HAM10000 datasets.

|         | PAD-UFES-20 | | | HIBA | | | HAM10000 | | |
|---------|------|------|------|------|------|------|------|------|------|
|         | Acc | BA | AUC | Acc | BA | AUC | Acc | BA | AUC |
| w/o p.s | 89.6 | 89.9 | .926 | 89.5 | 89.3 | .933 | 86.0 | **84.6** | .919 |
| w/ p.s  | **90.9** | **90.2** | .926 | **92.0** | **91.9** | **.954** | **87.3** | 83.5 | **.923** |

mance gains across all three target datasets (e.g., accuracy gains 89.6%→90.9% on PAD-UFES-20, 89.5%→92.0% on HIBA, 86.0%→87.3% on HAM10000). Hence, although global performance does not critically depend on positive sampling due to the dominance of benign signal, this strategy ensures the model encounters rare malignant phenotypes during pre-training, improving the model's ability to discriminate diverse lesion characteristics in downstream tasks.

## F.4 Image encoder finetuning

Restricting fine-tuning to the image embedding layers leads to improved results because it mitigates catastrophic forgetting. In fact, there is a significant domain shift between the reference and the target datasets, both because of the diverging nature of their metadata attributes and due to the different modality of the images in each dataset. By updating only the embedding layers, SLIMP preserves the representations learned on the much larger (and with richer metadata) SLICE-3D dataset while still adapting to the divergent charac-

Table 14: Comparison of full fine-tuning (✓) and embeddings-only tuning (✗) across target datasets. Best results in **bold**.

| Dataset | FT | Acc | BAcc | F1 | AUC |
|---------|-----|------|------|------|------|
| PAD-UFES-20 | ✓ | 87.0 | 86.7 | 0.883 | 0.922 |
|             | ✗ | **90.9** | **90.2** | **0.921** | **0.926** |
| HIBA | ✓ | 88.9 | 88.7 | 0.898 | 0.937 |
|      | ✗ | **92.0** | **91.9** | **0.925** | **0.954** |
| HAM10000 | ✓ | 86.5 | 73.7 | 0.606 | 0.917 |
|          | ✗ | **87.3** | **83.5** | **0.707** | **0.923** |

teristics of the target datasets. To validate this, we provide Table 14 comparing two scenarios. The first row of each dataset reports the performance after full fine-tuning of all encoder parameters, while the second one reports the strategy adopted in SLIMP, namely limiting the fine-tuning to the embedding layers only. We observe that the latter strategy consistently yields improved performance across all datasets and metrics.

## F.5 Patient batch size

We examine the impact of the patient batch size considered in the continual pre-training of the SLIMP on the PAD-UFES-20 dataset. Table 15 shows how the patient batch size affects performance on binary skin lesion classification. We observe that smaller batch sizes, such as $B = 4$ and $B = 8$, yield slightly lower Balanced Accuracy (BA) and F1 scores, while larger batch sizes lead to improved performance across all metrics but AUC. $B = 64$ achieves the highest BA of 90.2% and an F1 score of 0.921. Interestingly, further increasing the batch size (e.g., $B = 128$ or $B = 256$)

Table 15: Performance of the SLIMP method with different batch sizes (B) during the continual self-supervised learning stage on the PAD-UFES-20 dataset. Best results in **bold**.

|                    | Acc | BA | F1 | AUC |
|--------------------|------|------|------|------|
| SLIMP$_{B=4}$   | 90.0 | 86.4 | .886 | .907 |
| SLIMP$_{B=8}$   | 89.1 | 88.4 | .906 | .911 |
| SLIMP$_{B=32}$  | 88.7 | 88.4 | .898 | **.928** |
| SLIMP$_{B=64}$  | **90.9** | **90.2** | **.921** | .926 |
| SLIMP$_{B=128}$ | 89.6 | 89.1 | .908 | .918 |
| SLIMP$_{B=256}$ | 89.6 | 89.1 | .908 | .927 |

does not result in further performance gains and, in most cases, slightly decreases overall performance. This further highlights the importance of carefully choosing the patient batch size considered in the pre-training, as it can significantly impact performance. The choice $B = 64$ strikes an effective balance, justifying its choice as the reference configuration.

## F.6 PATIENT-DISJOINT SPLITS

To evaluate the robustness of SLIMP under stricter evaluation protocols, we additionally measure performance using patient-disjoint splits, where all lesions originating from the same patient are assigned to the same split. We construct these splits while maintaining a lesion-level ratio as close as possible to the standard 90%-10% train/validation division.

Table 16: Evaluation of SLIMP under standard (joint) vs. patient-disjoint evaluation splits. For the standard splitting strategy, we report the mean performance and corresponding standard deviation for each metric across three random dataset splits.

|  |  | Acc | BA | F1 | AUC |
|---|---|---|---|---|---|
| disjoint | PAD-UFES-20 | 89.5 | 89.6 | .894 | .940 |
|  | HIBA | 91.7 | 91.8 | .915 | .954 |
|  | HAM10000 | 87.9 | 80.8 | .676 | .926 |
| joint | PAD-UFES-20 | $90.6 \pm 0.9$ | $90.4 \pm 0.8$ | $.909 \pm .017$ | $.935 \pm .001$ |
|  | HIBA | $91.2 \pm 1.4$ | $91.2 \pm 1.4$ | $.911 \pm .016$ | $.951 \pm .010$ |
|  | HAM10000 | $87.8 \pm 0.4$ | $81.6 \pm 1.7$ | $.694 \pm .016$ | $.926 \pm .000$ |

The top three rows of Table 16 report the downstream classification performance considering these patient-disjoint splits across the PAD-UFES-20, HIBA, and HAM10000 datasets, while the last three rows of the table summarize the performance obtained using our standard lesion-level splitting strategy over three random seeds. We observe that the patient-disjoint results consistently fall within the variation ranges observed in the multi-seed lesion-level results. For example, HIBA achieves a Balanced Accuracy of 91.8% under patient-disjoint splits, which lies well within the range of 91.2% ± 1.4% obtained under our standard lesion-level splits. Similar trends are observed for PAD-UFES-20 and HAM10000. In several cases (e.g., HIBA and HAM10000), the patient-disjoint performance even surpasses the average performance of the lesion-level splits. These findings indicate that SLIMP does not rely on patient overlap across splits to achieve strong downstream performance, and that its representations remain stable under patient-level isolation.

## F.7 RETRIEVAL ABLATIONS

To further evaluate our retrieval-based metadata extrapolation approach, we consider comparison with two alternative strategies. The proposed approach shown in Figure 2 (right) operates in two stages. In the first stage image features from the target dataset are used to retrieve the closest lesion-level metadata from the reference dataset, and in the second stage the retrieved image and lesion embeddings are jointly used to retrieve the closest patient-level metadata. This ensures that the retrieved patient metadata is semantically aligned with the target lesion, rather than simply being the metadata associated with

Table 17: Comparison of three alternative metadata extrapolation strategies. Best results in **bold**.

| Dataset | Method | Acc | BAcc | AUC |
|---|---|---|---|---|
| PAD-UFES-20 | I→I | 73.0 | 73.5 | .765 |
|  | I→L | 73.5 | 73.2 | .765 |
|  | I→L→P | **77.0** | **77.0** | **.814** |
| HIBA | I→I | 77.2 | 77.2 | .815 |
|  | I→L | 76.5 | 76.6 | **.862** |
|  | I→L→P | **81.5** | **81.3** | .861 |
| HAM10000 | I→I | 83.3 | 67.5 | .830 |
|  | I→L | **83.4** | 63.6 | .833 |
|  | I→L→P | 82.2 | **71.0** | **.836** |

the initially retrieved reference lesion. Table 17 compares this approach with two alternatives. The first alternative, considers Image-to-Image (I→I) retrieval, where we retrieve the most similar lesion from the reference dataset relying solely on the image features and directly use the lesion and patient metadata associated with the corresponding lesion. In the second alternative (I→L), we consider a single-stage approach where lesion metadata are retrieved from the reference dataset given the image features, combined with the metadata of the patient who has the retrieved lesion. The results show that the full two-stage retrieval strategy (I→L→P) consistently yields the strongest overall performance across all datasets, highlighting the advantage of retrieving patient-level using the learned embedding space, rather than relying solely on image-based or lesion-level associations.

# G ADDITIONAL EXPERIMENTS

## G.1 KNN CLASSIFICATION PERFORMANCE

To further enhance our evaluation protocol, we performed k-nearest neighbors (kNN) classification for the downstream skin lesions classification task. Unlike linear probing, kNN offers a training-free evaluation that directly measures how well the learned feature space clusters samples of the same class. This protocol is widely adopted in contrastive and self-supervised learning, as it avoids introducing additional parameters or optimization choices while still reflecting the discriminative power of the representations. As reported in Table 18, SLIMP consistently surpasses all baselines across datasets, with the sole exception of HAM10000, and achieves an average accu-

Table 18: **kNN accuracy** (%) on the binary classification task across four target datasets. The average performance is reported in the last column. Best results are in **bold**, second best are underlined. (*PAD: PAD-UFES-20*)

| Method | PAD | HAM10000 | HIBA | PH2 | AVG |
|--------|-----|----------|------|-----|-----|
| MAE | 66.1 | 85.8 | 76.5 | **95.0** | 80.9 |
| BEiTv2 | 75.7 | 87.6 | 77.8 | 80.0 | 80.3 |
| DINOv2 | 72.6 | 83.8 | 77.2 | **95.0** | 82.2 |
| CLIP | 72.6 | 86.6 | 80.9 | **95.0** | 83.8 |
| SigLIP | 77.0 | 86.0 | 78.4 | 90.0 | 82.9 |
| SigLIP-2 | 75.7 | 85.1 | 80.9 | 90.0 | 82.9 |
| WL-CLIP | 76.5 | **89.7** | 85.2 | 90.0 | 85.4 |
| SimCLR | 67.4 | 87.2 | 80.3 | 62.5 | 74.4 |
| SLIMP$_{FLAT}$ | 81.3 | 84.1 | 77.8 | **95.0** | 84.6 |
| SLIMP$_{B=4}$ | **93.5** | 85.9 | **87.7** | **95.0** | **90.5** |

racy improvement of 5.1% over the second-best method. These results further support the findings reported in the main text, and demonstrate that the embedding space recovered by SLIMP is well-structured, even without task-specific fine-tuning.

## G.2 MULTICLASS CLASSIFICATION

In Table 19 we evaluate our proposed SLIMP method in a multiclass classification setting on PAD-UFES-20 dataset, in comparison with the baselines from Table 2. We report results for the overall Accuracy (Acc), F1-macro (which ensures equal contribution from minority classes), and F1-weighted (which accounts for class imbalance). Notably, SLIMP outperforms all baselines across all metrics, highlighting the robustness of SLIMP in handling imbalanced multiclass classification tasks. We note that techniques addressing class imbalance can be combined with SLIMP to further improve multiclass classification performance.

Table 19: Multiclass classification results on PAD-UFES-20 dataset. The *Metadata* column indicates whether metadata are used during the downstream classification task. Best results in **bold** second best are underlined.

| Method | Metadata | Acc | F1-macro | F1-weighted |
|--------|----------|-----|----------|-------------|
| MAE | ✗ | 70.0 | .631 | .692 |
| DINOv2 | ✗ | 73.0 | .614 | .726 |
| BEiTv2 | ✗ | 74.4 | .714 | .738 |
| CLIP | ✗ | 70.9 | .584 | .698 |
| SigLIP | ✗ | 73.9 | .680 | .724 |
| SigLIPv2 | ✗ | 74.8 | .700 | .745 |
| WL-CLIP | ✗ | 72.2 | .650 | .726 |
| SimCLR | ✗ | 84.2 | .688 | .826 |
| SBCL | ✗ | 45.7 | .289 | .433 |
| SLIMP | ✓ | **85.2** | **.833** | **.845** |
| TFormer | ✓ | 78.7 | .698 | .792 |

## G.3 RETRIEVAL

We conduct Image-to-Text (I2T) and Text-to-Image (T2I) downstream retrieval tasks across three target datasets (PAD-UFES-20, HAM10000, HIBA) comparing our proposed method, SLIMP with multi-modal baselines such as CLIP, SigLIP, SigLIP-2 and WhyLesionCLIP. For the baseline methods, we convert the tabular metadata into natural language descriptions using a large language model (GPT-4o). For SLIMP, both I2T and T2I tasks are performed using tabular metadata processed directly by our tabular encoder. The retrieval follows an instance-level protocol, where for T2I the ground truth is the lesion image described by a given description/metadata instance, and for I2T the true match is the specific set of either tabular metadata or textual description, corresponding to the input image. Queries for both tasks are drawn from the validation split of each target dataset, which remains unseen during all training phases.

Table 20: **Image-to-Text** retrieval performance on three target datasets. We compare SLIMP against cross-modal pretraining baselines; CLIP, SigLIP, SigLIP-2, and WhyLesionCLIP (WL-CLIP). Retrieval is evaluated using Recall at rank k (R@k), Normalized Discounted Cumulative Gain at k (N@k), and mean average precision (mAP). Best results in **bold**, second best are underlined.

| Models | R@5 | R@10 | R@15 | R@20 | R@100 | N@5 | N@10 | N@15 | N@20 | N@100 | mAP |
|---|---|---|---|---|---|---|---|---|---|---|---|
| *PAD-UFES-20* | | | | | | | | | | | |
| CLIPViT-B | 3.3 | 7.0 | 10.2 | 13.7 | 51.5 | 1.8 | 3.0 | 3.9 | 4.8 | 11.5 | 3.2 |
| SigLIPViT-B | 6.5 | 8.0 | 12.6 | 14.4 | 53.5 | 4.3 | 4.9 | 6.3 | 6.7 | 13.5 | 5.5 |
| SigLIP-2ViT-B | 7.4 | 9.8 | 11.7 | 13.3 | 49.4 | 5.0 | 5.8 | 6.4 | 6.7 | 13.2 | 5.6 |
| WL-CLIPViT-L | 2.6 | 6.1 | 11.3 | 12.4 | 52.0 | 1.3 | 2.5 | 3.8 | 4.1 | 11.1 | 3.0 |
| SLIMPViT-S | **9.0** | **14.8** | **19.0** | **28.2** | **77.2** | **5.6** | **7.5** | **8.8** | **11.2** | **20.9** | **7.9** |
| *HAM10000* | | | | | | | | | | | |
| CLIPViT-B | 0.6 | 1.0 | 1.4 | 2.1 | 10.9 | 0.5 | 0.7 | 0.8 | 1.0 | 3.6 | 1.9 |
| SigLIPViT-B | 1.0 | 1.5 | 2.2 | 2.9 | 12.0 | 0.9 | 1.1 | 1.4 | 1.6 | 4.4 | 2.4 |
| SigLIP-2ViT-B | 0.6 | 1.2 | 2.2 | 2.8 | 11.8 | 0.8 | 1.0 | 1.3 | 1.6 | 4.5 | 2.5 |
| WL-CLIPViT-L | **1.2** | **2.6** | **3.6** | **4.8** | **21.7** | **1.2** | 1.7 | 2.2 | 2.7 | 7.5 | 3.6 |
| SLIMPViT-S | 1.0 | 1.9 | 2.4 | 2.9 | 15.5 | 0.9 | **3.3** | **4.9** | **5.8** | **13.1** | **18.5** |
| *HIBA* | | | | | | | | | | | |
| CLIPViT-B | 3.9 | 7.4 | 10.5 | 13.6 | 66.4 | 2.6 | 3.9 | 4.6 | 5.5 | 15.6 | 4.8 |
| SigLIPViT-B | 3.5 | 8.9 | 15.1 | 20.7 | 72.4 | 3.5 | 5.7 | 7.4 | 8.9 | 18.7 | 6.6 |
| SigLIP-2ViT-B | 3.6 | 6.2 | 14.7 | 19.7 | 78.0 | 2.2 | 3.0 | 5.5 | 6.9 | 18.6 | 4.9 |
| WL-CLIPViT-L | **11.6** | 18.0 | 24.8 | **35.1** | **90.1** | 7.9 | 10.4 | 12.3 | 15.5 | 26.3 | 10.9 |
| SLIMPViT-S | 9.4 | **20.0** | **27.5** | 33.8 | 89.2 | **15.2** | **20.2** | **24.5** | **27.7** | **51.5** | **32.4** |

We report the retrieval results for I2T and T2I tasks, in tables 20 and 21 respectively. We evaluate retrieval using three metrics: Recall at rank k (R@k), Normalized Discounted Cumulative Gain (N@k) and mean Average Precision (mAP). N@k rewards relevant items appearing higher in the ranking and is a particularly critical metric in clinical evaluation tasks. Across all three target datasets, our approach substantially outperforms the baselines in most cases, often by large margins, despite being based on a ViT-S backbone while the competing methods were evaluated with larger ViT-B/L models. The gains we report in PAD-UFES-20 and HIBA, where rich patient- and lesion-level metadata are available, underscore the robustness of our method in leveraging structured clinical information. On HAM10000 dataset, our model still achieves the best retrieval quality in terms of NDCG. Notably, we outperform WhyLesionCLIP on the mAP metric, with gains of **+4.9**, **+14.9**, and **+21.5** for I2T retrieval on PAD-UFES-20, HAM10000, and HIBA, respectively, and **+3.6**, **+11.3**, and **+18.0** for T2I retrieval on the same datasets.

Table 21: **Text-to-Image** retrieval performance on three target datasets. We compare SLIMP against cross-modal pretraining baselines; CLIP, SigLIP, SigLIP-2, and WhyLesionCLIP (WL-CLIP). Retrieval is evaluated using Recall at rank k (R@k), Normalized Discounted Cumulative Gain at k (N@k), and mean average precision (mAP). Best results in **bold**, second best are underlined.

| Models | R@5 | R@10 | R@15 | R@20 | R@100 | N@5 | N@10 | N@15 | N@20 | N@100 | mAP |
|---|---|---|---|---|---|---|---|---|---|---|---|
| *PAD UFES 20* | | | | | | | | | | | |
| CLIPViT-B | 6.1 | 8.5 | 9.8 | 11.5 | 50.7 | 4.1 | 4.9 | 5.3 | 5.7 | 12.6 | 4.5 |
| SigLIPViT-B | 4.4 | 7.2 | 10.2 | 13.0 | 54.8 | 3.0 | 4.0 | 4.9 | 5.6 | 13.1 | 4.5 |
| SigLIP-2ViT-B | 5.7 | 8.9 | 10.2 | 13.7 | 48.9 | 3.7 | 4.9 | 5.2 | 6.1 | 12.5 | 4.9 |
| WL-CLIPViT-L | 3.9 | 6.5 | 8.5 | 10.2 | 45.7 | 3.1 | 3.6 | 4.2 | 4.6 | 11.0 | 4.0 |
| SLIMPViT-S | **8.7** | **16.1** | **26.1** | **30.0** | **78.3** | **6.7** | **10.4** | **13.5** | **14.5** | **22.3** | **7.6** |
| *HAM10000* | | | | | | | | | | | |
| CLIPViT-B | 0.7 | 1.2 | 1.6 | 1.8 | 9.4 | 1.5 | 2.0 | 2.2 | 2.4 | 7.4 | 1.3 |
| SigLIPViT-B | 1.2 | 2.2 | 2.4 | 3.3 | 13.6 | 2.5 | 4.3 | 4.6 | 5.4 | 9.9 | 1.9 |
| SigLIP-2ViT-B | 0.9 | 1.8 | 2.8 | 3.5 | 14.0 | 1.3 | 2.1 | 2.8 | 3.3 | 9.5 | 1.8 |
| WL-CLIPViT-L | **1.5** | **3.1** | **4.7** | **6.5** | **19.7** | 3.0 | 5.0 | 6.0 | 7.0 | 11.6 | 2.3 |
| SLIMPViT-S | 1.1 | 2.0 | 2.7 | 3.4 | 16.8 | **34.2** | **36.6** | **39.8** | **41.2** | **46.6** | **13.6** |
| *HIBA* | | | | | | | | | | | |
| CLIPViT-B | 2.5 | 7.1 | 11.3 | 13.8 | 65.8 | 1.2 | 3.6 | 5.0 | 5.6 | 15.8 | 3.5 |
| SigLIPViT-B | 2.5 | 6.5 | 11.1 | 17.3 | 77.9 | 2.0 | 3.6 | 4.5 | 6.0 | 18.4 | 4.0 |
| SigLIP-2ViT-B | 3.7 | 10.5 | 13.2 | 18.7 | 69.8 | 4.0 | 6.8 | 7.7 | 9.6 | 18.2 | 5.4 |
| WL-CLIPViT-L | 9.3 | 17.9 | 21.3 | 28.0 | 84.0 | 7.4 | 11.5 | 12.6 | 14.7 | 23.8 | 8.6 |
| SLIMPViT-S | **10.4** | **20.4** | **27.7** | **32.0** | **92.0** | **45.0** | **52.1** | **54.8** | **57.2** | **57.6** | **26.6** |

### G.4 Textual data

We reproduce a concept-based interpretability (CBI) method (Patrício et al., 2024), by adapting CLIP on the SLICE-3D dataset, considering a ViT-B/16 backbone architecture which offers optimal results. This methodology uses visual-language models for exploiting textual concepts for melanoma classification offering three different variants; (1) the *Baseline* approach, which directly applies CLIP, selecting the label that achieves the highest cosine similarity between the image and text embeddings, (2) the *CBM* approach, which introduces dermoscopic concepts and utilizes melanoma-specific coefficients to make predictions and (3) the *GPT-CBM* approach, which extends each dermoscopic concept introduced in CBM with multiple textual descriptions by querying it into ChatGPT.

In Table 22 we compare the performance of the above approaches, with our proposed SLIMP method, across three different target datasets, in a 'melanoma vs all' classification scenario. SLIMP is only adapted during linear probing while all pre-trained models on SLICE-3D dataset remain unchanged, highlighting the robustness of the learned representations. SLIMP consistently outperforms all other approaches without the need of task-specific pre-training.

Table 22: Comparison of SLIMP method with CBI variants across three target datasets. Results for the proposed SLIMP method are obtained using a linear probing setting. Best results in **bold**.

| | PAD-UFES-20 | | | | HIBA | | | | HAM10000 | | | |
|---|---|---|---|---|---|---|---|---|---|---|---|---|
| | Acc | BA | F1 | AUC | Acc | BA | F1 | AUC | Acc | BA | F1 | AUC |
| Baseline | 23.9 | 51.3 | .044 | .422 | 68.5 | 54.8 | .261 | .502 | 72.0 | 58.6 | .247 | .595 |
| CBM | 78.7 | 69.6 | .109 | .778 | 48.2 | 61.3 | .333 | .659 | 54.1 | 58.8 | .238 | .565 |
| GPT-CBM | 35.7 | 57.3 | .051 | .599 | 48.8 | 61.7 | .336 | .638 | 55.5 | 57.6 | .231 | .581 |
| SLIMP | **98.7** | **70.0** | **.571** | **.993** | **90.1** | **72.3** | **.600** | **.939** | **89.1** | **67.9** | **.452** | **.892** |

## H  Feature importance

In Figure 5 we estimate feature importance scores from the **last-layer self-attention maps** of the tabular transformer. Each attention matrix $A \in \mathbb{R}^{T \times T}$, with T the number of tokens ([cls] + features), is the standard dot product of queries and keys followed by a softmax activation function. We discard the [cls] token, as our downstream tasks rely on the global average pooling (GAP) of the output feature tokens coming from TRACE rather than the [cls] representation. After masking the diagonal and renormalizing each row, the normalized importance of feature $j$ is computed as

$$\text{Imp}_j = \frac{\mathbb{E}\left[\frac{1}{T-1}\sum_{i \neq j} \frac{A_{ij}}{\sum_{k \neq j} A_{ik}}\right]}{\sum_m \mathbb{E}\left[\frac{1}{T-1}\sum_{i \neq m} \frac{A_{im}}{\sum_{k \neq m} A_{ik}}\right]}, \quad \sum_j \text{Imp}_j = 1,$$

where $i$ indexes querying features, $j$ receiving feature and $k$ runs over all possible receivers in row $i$. The resulting distributions in Figure 5 highlight which patient- and lesion-level features dominate the model's internal attention mechanism. We observe that age, the number of lesions per patient and the Fitzpatrick skin type (where available) consistently dominate the outer level of the architecture, reflecting their strong influence in clinical diagnosis. Importantly, these features are considered among the most relevant according to the dermatology literature. In addition, for the PAD-UFES-20 dataset the inner tabular transformer attends strongly to critical features such as the anatomical region of the lesion and indicators of lesion change detection (e.g., whether the lesion has grown or itched). For HIBA and SLICE-3D, we observe a similar pattern; morphology, size and localization systematically receive higher attention by our lesion-level descriptors, suggesting that SLIMP consistently focuses on clinically meaningful attributes at both hierarchical levels.

## I  Qualitative assessment

Figure 6 shows the t-SNE (Hinton & Roweis, 2002) embeddings of the three SLIMP variants presented in Table 4, on the PAD-UFES-20 dataset. We observe a better separation between benign and malignant lesions when metadata are considered during pre-training.

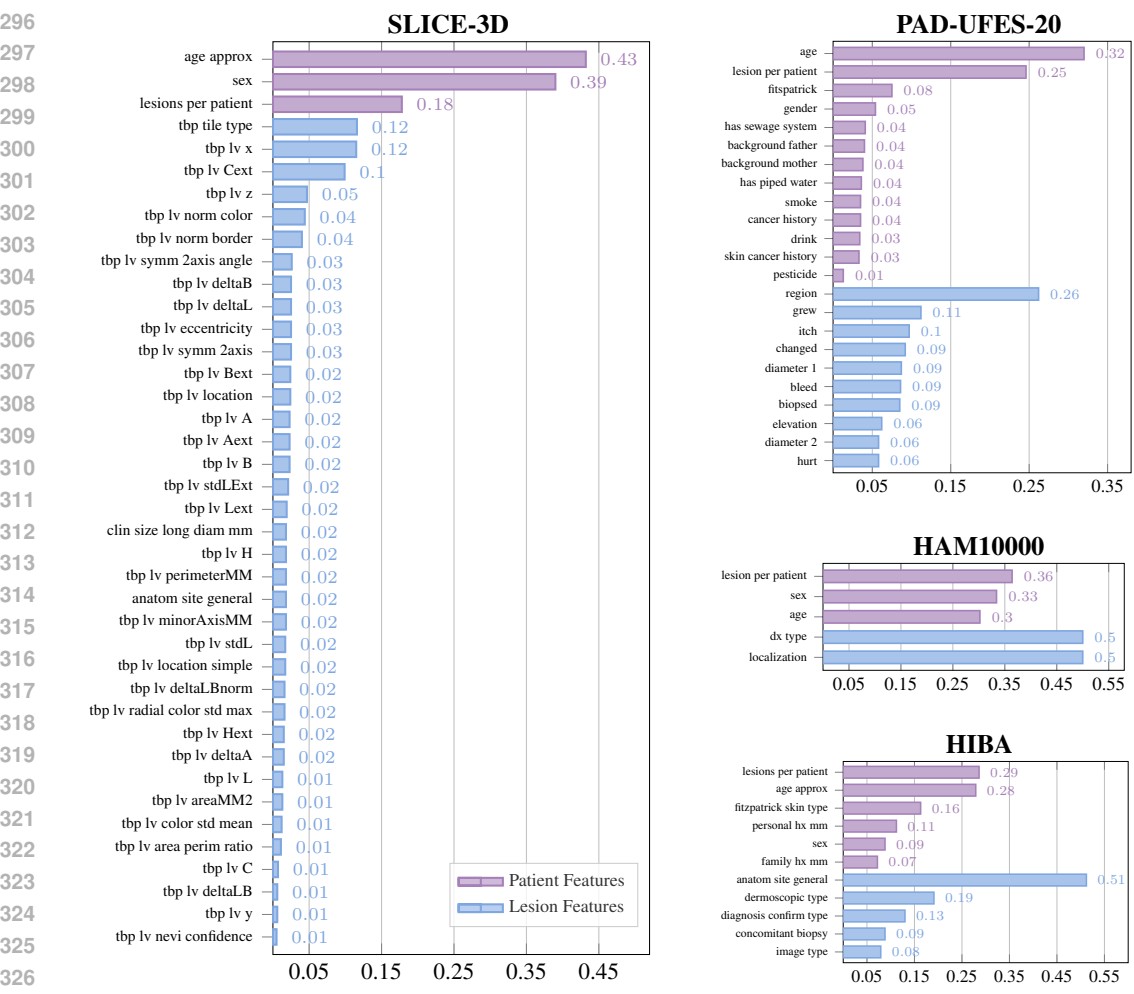

Figure 5: Normalized feature importance scores for patient-level and lesion-level features. The importance scores are derived from the attention mechanism of each tabular transformer respectively.

Figure 7 provides a qualitative evaluation of the proposed metadata retrieval process. For each target sample (left), we display the image corresponding to the lesion metadata (right) retrieved from the SLICE-3D dataset. Although our method does not retrieve images but rather lesion metadata, the images corresponding to the retrieved metadata exhibit notable visual similarity compared to the input image in terms of lesion morphology, color, and overall structure, even under challenging conditions

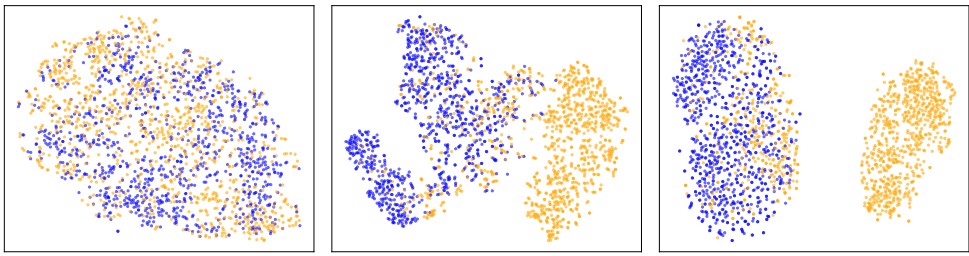

Figure 6: t-SNE visualization of SLIMP features for benign and malignant lesions in the PAD-UFES-20 dataset. **Left:** Pre-training using image encoder alone; **Middle:** Pre-training using image and lesion metadata; **Right:** Pre-training using images with lesion and patient-level metadata.

such as hair occlusion, presence of artifacts, and differing imaging modalities. This further supports the semantic consistency captured by our learned representations and validates the effectiveness of the retrieval-based metadata extrapolation strategy.

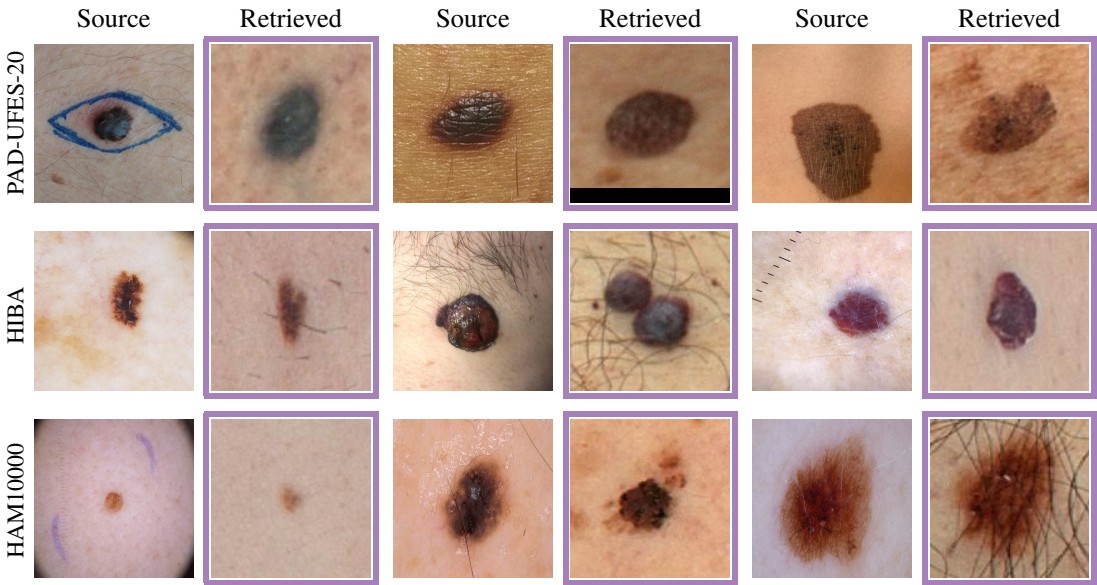

Figure 7: Qualitative examples of our retrieval-based metadata extrapolation method. For each lesion image (left) from the target dataset, we display the image associated with the lesion-level metadata retrieved by our model (right) from the reference dataset (SLICE-3D).

Figures 8 and 9 presents randomly selected lesions from each dataset validation split, with the corresponding attention maps extracted from the pre-trained image encoders of MAE, BEiTv2, DINOv2, CLIP, WL-CLIP, SimCLR and SLIMP (ours) in this order. We note that SLIMP effectively localizes the majority of the lesions, regardless of differences in lesion shape, texture and color. This consistency in identifying relevant lesion regions indicates the robustness of the learned representations across diverse datasets that exhibit a high variation in visual appearance, also due to different imaging modalities. It also showcases the ability of the model to focus on relevant skin-lesion features, supporting the improved downstream classification performance, and suggesting that the method can enhance the interpretability and reliability of the results.

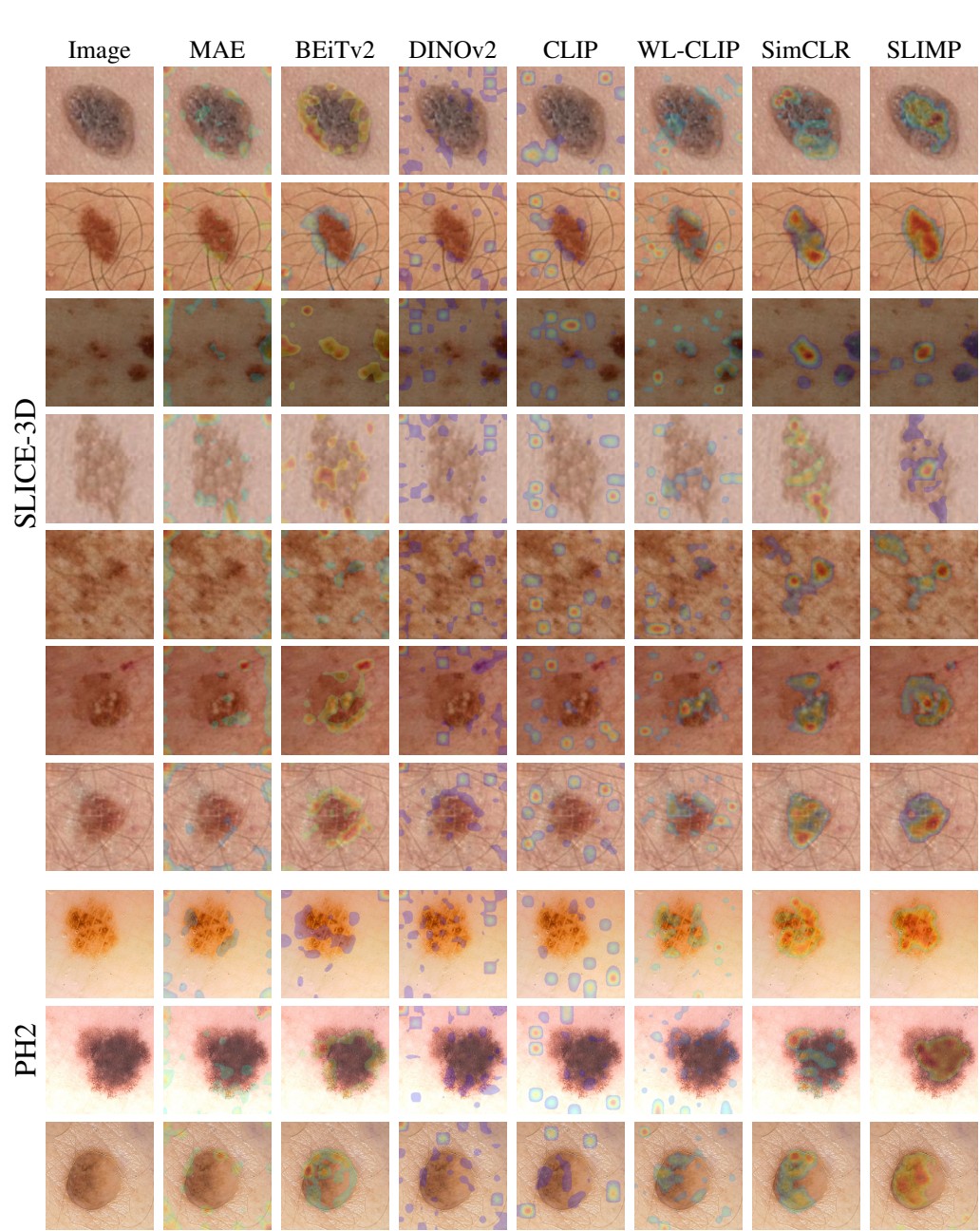

Figure 8: Attention maps obtained from the last self-attention block of the image encoder across different pre-trained models. The leftmost column shows the original image, while the remaining columns display heatmap overlays from MAE, BEiTv2, DINOv2, CLIP, WL-CLIP, SimCLR, and our proposed SLIMP (rightmost column). The top seven rows correspond to samples from SLICE-3D reference dataset, while the bottom three rows correspond to samples from PH2 target dataset.

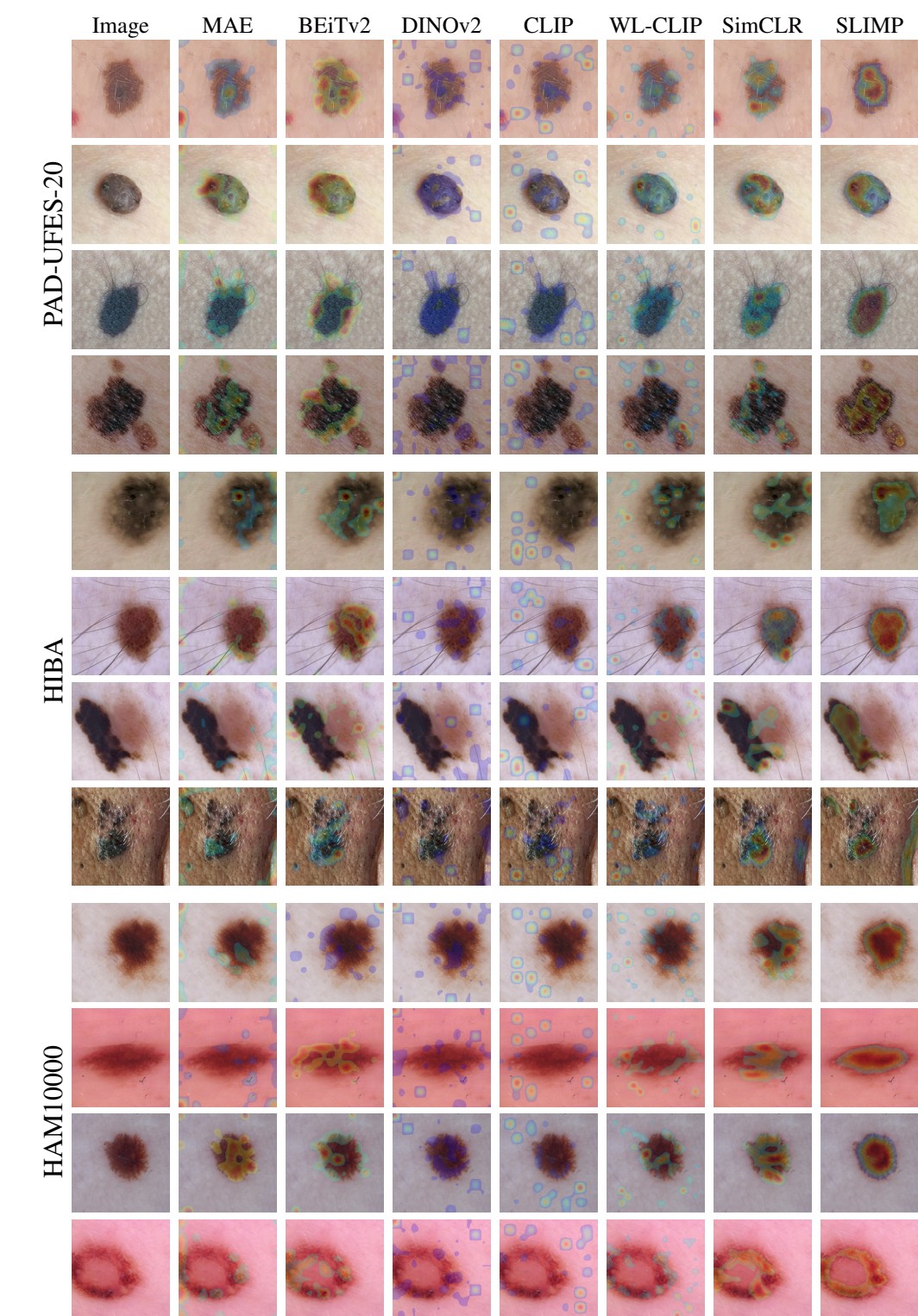

Figure 9: Attention maps obtained from the last self-attention block of the image encoder across different pre-trained models. The leftmost column shows the original image, while the remaining columns display heatmap overlays from MAE, BEiTv2, DINOv2, CLIP, WL-CLIP, SimCLR, and our proposed SLIMP (rightmost column). The top four rows correspond to samples from PAD-UFES-20 target dataset, the middle four rows correspond to samples from HIBA target dataset, and the bottom four rows correspond to samples from HAM10000 target dataset.

