# OpenReview forum: "Skin Lesion Phenotyping via Nested Multi-modal Contrastive Learning"
_ICLR.cc/2026/Conference — Submitted to ICLR 2026_

### Official Review · Reviewer_4Q9R · 2025-10-28

**Soundness:** 2
**Presentation:** 1
**Contribution:** 2
**Rating:** 2
**Confidence:** 4

**Summary:**

This paper presents SLIMP (Skin Lesion Image–Metadata Pre-training), a multi-modal self-supervised framework combining lesion images, lesion-level metadata, and patient-level metadata. It introduces a two-level (“nested”) contrastive learning scheme and extends it with (1) continual pre-training for addressing differences in metadata attributes and imaging modalities and (2) retrieval-based metadata extrapolation to handle missing modalities. Experiments on multiple dermatology datasets show consistent improvements over self-supervised and multi-modal baselines.

**Strengths:**

1. The retrieval-based approach for getting pseudo-metadata is an interesting idea for coping with missing modalities in clinical datasets.
2. Experiments are comprehensive, covering multiple datasets, ablations, and low-shot settings.

**Weaknesses:**

1. The key idea, “Nested Contrastive Multi-Modal Learning,” is not well-defined. The method essentially applies two standard InfoNCE losses at the lesion and patient levels. The “nested” terminology seems to only indicate the two-level setup rather than a novel framewrok.
2. Related work lacks discussion on dermatology-specific foundation models (e.g., PanDerm[1]).
[1] Yan, S., Yu, Z., Primiero, C., Vico-Alonso, C., Wang, Z., Yang, L., ... & Ge, Z. (2025). A multimodal vision foundation model for clinical dermatology. Nature Medicine, 1-12.
3. The methodological section is logically disorganized. The description merges multiple concepts—nested contrastive learning, continual pre-training, and retrieval-based extrapolation—without clear separation between training phases, losses, and objectives. e.g.  The handling of catastrophic forgetting in Sec. 3.2 is reduced to a hand-wavy “fine-tune only a restricted set of parameters,” without specifying which.
The methodology is not yet publication-ready and requires substantial rewriting to clarify the model pipeline and training logic.

**Questions:**

1. In Table 2, what exactly differentiates supervised learning (TFormer) from continual pre-training (SLIMPImage) in terms of training data, loss functions, and supervision signals? How should readers interpret “continual pre-training” as a self-supervised adaptation step without reference labels? In addition, please clarify the relationship between continual pre-training and linear probing.
2. Could the authors provide a clear process of training stages (pre-training → continual pre-training) and specify which datasets and losses are used at each step? e.g., The statement like "For continual pre-training on target datasets, we fine-tune the embedding layers of the image and metadata encoders, keeping their attention layers frozen." is confusing.
3. In the retrieval-based metadata extrapolation, how is retrieval quality validated? Could mismatched metadata harm classification performance or introduce bias?
4. Regarding comparisons with generic and multi-modal foundation models (e.g., MAE, CLIP, SigLIP) — were those baselines also pre-trained on skin lesion data? If not, this comparison may be unfair since SLIMP benefits from in-domain pre-training. Please clarify how the domain mismatch was handled.

---

> ### Author Response · Authors · 2025-11-26
> **Answer to Reviewer 4Q9R (Part 1)**
>
> We sincerely thank the reviewer for the constructive feedback and for highlighting the strengths of our work. In particular, we appreciate the recognition of our retrieval-based strategy for generating pseudo-metadata, which offers a practical and effective solution for handling missing modalities in real-world clinical datasets. We are also grateful for the reviewer’s acknowledgment of the breadth and depth of our experimental study, including evaluations across multiple datasets, detailed ablations, and challenging low-shot scenarios. These insights are highly valuable, and we address the reviewer’s remaining concerns below.
>
> ### **Weakness 1: Limited novelty. Main contribution seems to be standard InfoNCE**
>
> [Answer reiterated also in: Reviewer b9ns (Part 2) and Reviewer CbJ2 (Part 1)]
>
> We agree with the reviewer that SLIMP uses the standard InfoNCE loss. The crucial difference and innovation lie not in modifying the loss formula but in the novel nested multi-modal scheme through which the standard loss is architecturally applied.
>
> **Nested architecture vs. flat loss:**
> Unlike standard contrastive learning, which operates on a single flat level (e.g., image–image or image–text pairs), or flatly combined modalities (as in $\text{SLIMP}_\text{FLAT}$), SLIMP introduces a two-level nested contrastive framework specifically designed to model the **compositional clinical hierarchy** of skin lesion data:
>
> - The **inner level** ($L_{lesions}$) maximizes the agreement between the lesion image ($w_p^l$) and the corresponding lesion-specific metadata ($h_l^p$) to obtain fine-grained lesion representations.
> - The **outer level** ($L_{patient}$) aggregates the lesion-level representations into a  patient phenotype summary ($z_p$) and then contrasts this summary with the patient-level metadata.
>
> This hierarchical and multi-modal structure explicitly encourages learning representations that incorporate patient-level context, leading to stronger, more generalizable features. To the best  of our knowledge, this structure is novel and fundamentally different from standard InfoNCE pipelines, which do not model nested, multi-modal clinical hierarchies.
>
> **Distinction from other hierarchical methods:**
> Moreover, the nested structure in SLIMP is fundamentally distinct from other hierarchical contrastive learning approaches. Compared to the works of Zhang et al. (2022) and Jiang et al. (2023) which enforce consistency across a **taxonomic or label hierarchy** (e.g., class lineage), SLIMP models a compositional hierarchy based on a relation of belonging (multiple lesions belong to a single patient). At the same time, our scheme spans multiple data modalities (image, lesion metadata, patient metadata) across its levels, contrasting with intra-modal hierarchical approaches like Wang et al. (2023), which apply contrastive loss across multiple levels of the same modality (as for example medical time-series).
>
> The effectiveness of this nested architecture is confirmed by our ablation study in Table 2. Collapsing our nested design into a single-level contrastive loss ($\text{SLIMP}_{\text{FLAT}}$) leads to substantial drops in performance (e.g., −4.9% BAcc loss on PAD-UFES-20 and −7.4% BAcc loss on HIBA) compared to the full nested SLIMP model. This clearly shows that the nested contrastive architecture, rather than a modified loss formula, is the crucial component driving SLIMP’s state-of-the-art gains.

---

> ### Author Response · Authors · 2025-11-26
> **Answer to Reviewer 4Q9R (Part 2)**
>
> ### **Weakness 2: Related work lacks discussion on dermatology-specific foundation models**
>
> We thank the reviewer for pointing out PanDerm (Yan et al., 2025) an important recent multimodal foundation model. We will include a detailed discussion of it in the revised Section 2. However, **PanDerm** focuses exclusively on **image-based modeling** and does not incorporate or contrast structured lesion-level or patient-level metadata, nor does it model the hierarchical lesion→patient relationships that SLIMP is designed for.
>
> **Methodological distinction:**
> While PanDerm leverages large-scale pre-training for image-text alignment and visual question answering, it does not explicitly model the structured, hierarchical metadata (lesion characteristics nested within patient history) that are central to clinical workflows. SLIMP by design exploits these structured dependencies, offering a complementary approach to text-based foundation models.
>
> **Evaluation:**
> To address this comment, we evaluated PanDerm-Base and PanDerm-Large on our exact splits for PAD-UFES-20, HIBA, and HAM10000. As shown in the table below (which will be added to the revised manuscript), **SLIMP**, while based on a much smaller ViT-Small backbone, **consistently outperforms** the significantly larger PanDerm-Large model. Notably, in terms of Balanced Accuracy, SLIMP compared to **PanDerm-Large** achieves +10.4% gain on PAD-UFES-20 (90.5% over 80.1%), +4.7% gain on HIBA (92.4% over 87.7%) and +6.9% gain on HAM10000 (84.5% over 77.6%).
> These results confirm that incorporating structured patient- and lesion-level metadata via our nested contrastive framework yields superior diagnostic performance compared to broad vision-language pre-training alone.
>
> |                        | **PAD** |        |           |       | **HIBA** |        |           |       | **HAM** |        |           |       |
> |------------------------|---------|--------|-----------|-------|----------|--------|-----------|-------|---------|--------|-----------|-------|
> |                               | Acc     | BAcc   | F1        | AUC   | Acc      | BAcc   | F1        | AUC   | Acc     | BAcc   | F1        | AUC   |
> | PanDerm-Base                  | 75.2    | 75.5   | 0.753     | 0.857 | 86.4     | 86.5   | 0.864     | 0.932 | 85.1    | 70.0   | 0.839     | 0.892 |
> | PanDerm-Large                 | 80.0    | 80.1   | 0.800     | 0.904 | 87.7     | 87.7   | 0.877     | 0.941 | 88.5    | 77.6   | 0.879     | 0.925 |
> | $\text{SLIMP}_{Image}$        | 76.5    | 76.4   | 0.784     | 0.816 | 80.2     | 80.3   | 0.807     | 0.872 | 85.9    | 71.1   | 0.566     | 0.875 |
> | $\text{SLIMP}_{Image+Lesion}$ | 90.0    | 89.5   | 0.912     | 0.918 | 87.7     | 87.5   | 0.886     | 0.897 | 87.0    | 78.6   | 0.663     | 0.909 |
> | $\text{SLIMP}_{B=4}$          | 90.9    | 90.2   | 0.921     | 0.926 | 92.0     | 91.9   | 0.925     | 0.954 | 87.3    | 83.5   | 0.707     | 0.923 |
> | $\text{SLIMP}_{B=4}$          | 90.9    | 90.5   | 0.919     | 0.929 | 92.6     | 92.4   | 0.931     | 0.944 | 87.6    | 84.5   | 0.717     | 0.929 |
>
> ---
>
> ### **Question 1: TFormer and SLIMP training clarifications**
>
> We are happy to clarify the distinctions in Table 2 regarding training data, supervision signals, and loss functions. We provide here a structured comparison for improved clarity:
>
> **SLIMP (Continual Pre-training):** This is a **Self-Supervised Domain Adaptation** step.
> - **Data:** It uses images and metadata from the target dataset but **zero class labels**.
> - **Objective:** The model minimizes the **InfoNCE loss** at both levels (lesion and patient) to align the target dataset's specific metadata structure and image modality with the pre-trained SLICE-3D feature space.
> - **Mechanism:** As detailed in our methodology, this step is totally class-agnostic. We fine-tune the embedding layers to adapt to the new distribution while keeping the attention blocks frozen.
>
> **Linear Probing (Evaluation):** This follows the Continual Pre-training step.
> - **Data:** Uses the target dataset images/metadata and the class labels (training split).
> - **Objective:** The encoder is **frozen**. We train only a single linear classifier on top of the representations using **Cross-Entropy loss**. This is the standard protocol to evaluate the quality of the self-supervised features learned during the continual pre-training stage.
>
> **TFormer (Supervised Baseline):**
> - **Data:** Uses images, metadata, and **ground-truth class labels** from the start.
> - **Objective:** This model is trained **fully supervised** in an end-to-end manner on the target dataset.
> - **Contrast:** Unlike SLIMP, TFormer relies entirely on supervised signals and does not perform any self-supervised adaptation or freezing of parameters. It represents the fully supervised upper bound for that architecture.

---

> ### Author Response · Authors · 2025-11-26
> **Answer to Reviewer 4Q9R (Part 3)**
>
> ### **Question 2: Methodology structure**
>
> **Methodology structure:**
> We appreciate the reviewer’s constructive comment regarding the structure of our manuscript and agree that the current presentation of Section 3 can be made clearer. In the revised manuscript, **Section 3 will be reorganized into two explicit parts to eliminate ambiguity**. The first part will address a unified **pre-training section**. We will merge the description of reference pre-training (SLICE-3D) and continual pre-training (target datasets) into a single logical block, as both represent the self-supervised representation learning phase. The second part will present **retrieval-based metadata extrapolation** and the **downstream evaluation tasks**, mirroring the flow of Figures 1 and 2.
>
> **Pipeline protocol:**
> To address the reviewer's request for a clear process of training stages, datasets, and losses, the exact protocol is as follows:
>
> **Stage 1: Reference Pre-training (Self-Supervised)**
> - Dataset: SLICE-3D (Reference).
> - Objective: Nested lesion-level and patient-level InfoNCE losses (Eqs. 1–2).
> - Optimization: All modules are fully updated: the ViT backbone, the lesion metadata encoder, and the patient metadata encoder.
>
> **Stage 2: Continual Pre-training (Self-Supervised Adaptation)**
> - **Dataset:** Target Dataset (e.g., PAD-UFES-20, HIBA, HAM10000, PH2) without labels.
> - **Objective:** The same nested lesion- and patient-level InfoNCE objectives.
> - **Optimization:** To address the concern about "hand-wavy" descriptions, we specify that only the embedding layers are updated (specifically the ViT patch embedding layer and the input embedding layers of the TRACE metadata transformer). All ViT Transformer blocks and TRACE Transformer blocks remain frozen. This acts as a lightweight domain adaptation mechanism . As shown in Table 13, this strategy consistently outperforms full fine-tuning.
>
> **Stage 3: Downstream Evaluation (Supervised)**
> - **Dataset:** Target Dataset with labels.
> - **Protocol:** Linear Probing. All encoders (Image and Metadata) remain **frozen**.
> - **Objective:** A linear classifier is trained using Binary Cross-Entropy (BCE) in the case of  benign vs. malignant classification or Cross-Entropy (CE) for multi-class classification.
> - **Missing Metadata:** If the target lacks metadata, we first apply the **retrieval-based metadata extrapolation** to borrow metadata from the SLICE-3D embedding space, then perform linear probing. Retrieval is strictly an inference-time operation and does not modify the encoders.
>
> ---
>
> ### **Question 3: Metadata extrapolation**
>
> We thank the reviewer for this crucial question regarding the reliability of the extrapolation module. We validate retrieval quality through three complementary analyses.
>
> **Validation of retrieval quality:** Retrieval quality is validated extensively in **Appendix E**, where we analyze the structure and separability of the joint embedding space. **Table 6** shows a large separation between the cosine-similarity distributions of matching vs. non-matching metadata pairs on SLICE-3D (median 0.836 vs. −0.006), demonstrating that the embedding space is well-structured and retrieval is highly reliable. Additionally, **Figure 4** compares the similarity distributions of (i) true metadata, (ii) retrieved metadata from SLICE-3D, and (iii) retrieved metadata for each target dataset. Retrieved metadata closely match the true distribution on SLICE-3D, and even across target datasets with strong domain shift (PAD-UFES-20), the distributions remain well aligned, indicating stable cross-domain retrieval.
>
> **Direct Correctness:** To directly measure correctness, **Table 7** reports Recall@k for retrieving the ground-truth metadata on a held-out SLICE-3D validation set. The model achieves R@1 = 45.1%, R@5 = 76.6%, and R@100 = 99.1%, showing that the correct metadata almost always appears among the top retrieval candidates. This demonstrates that metadata extrapolation is statistically reliable and does not produce arbitrary mismatches.
>
> **Qualitative Check:** We further performed a visual inspection of images corresponding to retrieved metadata, observing that they largely share phenotypic characteristics (e.g., size, border, color) with the query images.
>
> **Impact of Mismatched Metadata (Harm vs. Benefit):**
> If metadata extrapolation were harmful, we would expect it to reduce downstream performance. Instead, **Table 2** shows that Ret-SLIMP consistently outperforms Pre-SLIMP (which uses no metadata) across all datasets, even in the most challenging settings where metadata schemas differ substantially, as in PAD-UFES-20 (smartphone images) and HIBA (dermoscopy). This empirical evidence demonstrates that while retrieval-based metadata extrapolation is not perfect, it does not introduce harmful bias. On the contrary, it provides a consistent and measurable benefit to representation learning.

---

> ### Author Response · Authors · 2025-11-26
> **Answer to Reviewer 4Q9R (Part 4 - final part)**
>
> ### **Question 4: Baseline Fairness**
> We believe the comparison is fair, robust, and informative. We address the concern regarding domain mismatch by including three distinct categories of baselines in Table 2, each serving a specific scientific purpose:
>
> 1. **Generic Foundation Models (MAE, CLIP, SigLIP, DINOv2):** The reviewer is correct that these models were not pre-trained on skin lesion data. We include them specifically to demonstrate a critical finding: **domain-specific pre-training** (SLIMP) yields superior representations compared to **massive-scale generic pre-training**, even when the generic models use significantly larger backbones (e.g., ViT-B or ViT-L vs. SLIMP's ViT-S) . This is a standard benchmark in medical imaging to justify the necessity of domain adaptation over simply applying off-the-shelf foundation models. Regarding evaluation consistency, we ensured procedural fairness by evaluating all generic models under the same supervised linear-probing protocol. None were used in zero-shot mode, each receives the same supervised signal on the target dataset during linear probing.
> 2. **Controlled In-Domain Baseline (SimCLR):** To strictly isolate the benefit of our method from the benefit of the data, we compared Pre-SLIMP against **SimCLR pre-trained on the exact same SLICE-3D dataset**. Since both models saw the exact same skin lesion images during pre-training, any performance difference is strictly due to the **SLIMP methodology** (nested multi-modal contrastive learning) rather than domain exposure. Pre-SLIMP significantly outperforms this in-domain SimCLR baseline (e.g., **+6.1%** Balanced Accuracy on PAD-UFES-20), proving that our architectural contribution provides value beyond simple in-domain pre-training.
> 3. **Dermatology-Specific Baselines:** To further validate that our performance gains stem from the proposed methodology rather than just domain exposure, we evaluated against state-of-the-art models explicitly designed for skin lesion analysis. We compared SLIMP with WhyLesionCLIP (WL-CLIP), a foundation model explicitly fine-tuned on biomedical image-text pairs, and SBCL, a contrastive learning method tailored for long-tailed dermatology datasets. SLIMP outperforms both, suggesting that our integration of structured tabular metadata provides superior guidance compared to unstructured text or specialized sampling strategies alone. Furthermore, we compared our method against TFormer, a fully supervised model trained end-to-end on the target dataset labels. Notably, SLIMP matches or exceeds this fully supervised upper bound on most datasets, demonstrating the high quality of our self-supervised representations.
>
> **Protocol Adherence:**
> Finally, we maintained a strict Linear Probing protocol (freezing the encoder) across all experiments. This is the standard method in self-supervised learning literature to isolate the quality of the pre-trained representations. Fine-tuning the baselines' encoders would deviate from this protocol, making it difficult to disentangle the quality of the initial features from the plasticity of the model during adaptation.

---

### Official Review · Reviewer_CbJ2 · 2025-10-31

**Soundness:** 4
**Presentation:** 4
**Contribution:** 2
**Rating:** 4
**Confidence:** 5

**Summary:**

The paper introduces SLIMP (Skin Lesion Image-Metadata Pre-training), a novel nested multi-modal contrastive learning approach for learning rich representations of skin lesions. The method combines lesion images with both lesion-level and patient-level metadata to improve performance on downstream skin lesion classification tasks. The authors claim that by capturing complementary information from different modalities, SLIMP produces more representative and generalizable features. They also propose methods for adapting the model to target datasets with different metadata structures, including a continual pre-training approach and a metadata extrapolation strategy for image-only datasets.

**Strengths:**

1.  The paper addresses an important problem in medical imaging—the effective use of multi-modal data for diagnosis. Improving skin lesion classification can have a direct impact on early melanoma detection.
2.  The overall approach is presented clearly, and the motivation is well-explained. Figure 1 provides a good visual summary of the SLIMP architecture.
3.  The paper considers the practical challenges of working with multiple medical datasets, such as diverging metadata schemas, and proposes reasonable solutions (continual pre-training and metadata extrapolation).

**Weaknesses:**

1.  The novelty of SLIMP appears to be rather limited. In Figure 1, the most crucial contribution seems to be the InfoNCE loss, which was proposed in 2018. Additionally, the feature concatenation approach is quite common.
2.  The authors claim several contributions (nested loss, continual pre-training, metadata extrapolation). However, there are no ablation studies to quantify the individual impact of each component. For example, how does the nested loss compare to a "flat" contrastive loss that combines all metadata? How much does continual pre-training improve performance compared to zero-shot transfer?
3.  The idea of transferring metadata based on image similarity is interesting but potentially risky. The paper should provide a more in-depth analysis of when this works and when it might fail (e.g., if visually similar lesions have very different metadata due to patient history).

**Questions:**

1.  Could you please provide ablation studies to demonstrate the individual contributions of the nested contrastive loss, the continual pre-training, and the metadata extrapolation? For instance, what is the performance if you only use a single-level contrastive loss?
2.  Regarding the metadata extrapolation, have you analyzed the failure cases? Are there instances where visually similar images from the reference and target datasets have significantly different (and clinically important) metadata, leading to incorrect "extrapolation"?

---

> ### Author Response · Authors · 2025-11-26
> **Answer to Reviewer CbJ2 (Part 1)**
>
> We sincerely thank the reviewer for their thorough assessment and recognition of our paper's strengths. We are pleased the reviewer finds the approach sound and well-presented, giving a score of excellent for soundness and presentation. We particularly appreciate the recognition that SLIMP addresses an important problem in medical imaging by enabling the effective use of multi-modal data for diagnosis, and that we propose reasonable solutions to overcome the practical challenges of working with multiple medical datasets and diverging metadata schemas.
>
> The reviewer raises crucial questions regarding the novelty, the quantification of contributions, and the analysis of extrapolation risks. We address each of these points below by providing further detail and clarifying the existing evidence within the manuscript.
>
> ### **Weakness 1: Limited novelty. Main contribution seems to be standard InfoNCE**
>
> [Answer reiterated also in: Reviewer b9ns (Part 2) and Reviewer 4Q9R (Part 1)]
>
> We agree with the reviewer that SLIMP uses the standard InfoNCE loss. The crucial difference and innovation lie not in modifying the loss formula but in the novel nested multi-modal scheme through which the standard loss is architecturally applied.
>
> **Nested architecture vs. flat loss:**
> Unlike standard contrastive learning, which operates on a single flat level (e.g., image–image or image–text pairs), or flatly combined modalities (as in $\text{SLIMP}_\text{FLAT}$), SLIMP introduces a two-level nested contrastive framework specifically designed to model the **compositional clinical hierarchy** of skin lesion data:
>
> - The **inner level** ($L_{lesions}$) maximizes the agreement between the lesion image ($w_p^l$) and the corresponding lesion-specific metadata ($h_l^p$) to obtain fine-grained lesion representations.
> - The **outer level** ($L_{patient}$) aggregates the lesion-level representations into a  patient phenotype summary ($z_p$) and then contrasts this summary with the patient-level metadata.
>
> This hierarchical and multi-modal structure explicitly encourages learning representations that incorporate patient-level context, leading to stronger, more generalizable features. To the best  of our knowledge, this structure is novel and fundamentally different from standard InfoNCE pipelines, which do not model nested, multi-modal clinical hierarchies.
>
> **Distinction from other hierarchical methods:**
> Moreover, the nested structure in SLIMP is fundamentally distinct from other hierarchical contrastive learning approaches. Compared to the works of Zhang et al. (2022) and Jiang et al. (2023) which enforce consistency across a **taxonomic or label hierarchy** (e.g., class lineage), SLIMP models a compositional hierarchy based on a relation of belonging (multiple lesions belong to a single patient). At the same time, our scheme spans multiple data modalities (image, lesion metadata, patient metadata) across its levels, contrasting with intra-modal hierarchical approaches like Wang et al. (2023), which apply contrastive loss across multiple levels of the same modality (as for example medical time-series).
>
> The effectiveness of this nested architecture is confirmed by our ablation study in Table 2. Collapsing our nested design into a single-level contrastive loss ($\text{SLIMP}_{\text{FLAT}}$) leads to substantial drops in performance (e.g., −4.9% BAcc loss on PAD-UFES-20 and −7.4% BAcc loss on HIBA) compared to the full nested SLIMP model. This clearly shows that the nested contrastive architecture, rather than a modified loss formula, is the crucial component driving SLIMP’s state-of-the-art gains.

---

> ### Author Response · Authors · 2025-11-26
> **Answer to Reviewer CbJ2 (Part 2)**
>
> ### **Weakness 2/Question 1: Missing ablations for nested loss, continual pre-training, and metadata extrapolation**
>
> We thank the reviewer for this comment and would like to kindly point out that all three requested ablations are included in the manuscript. Specifically, Tables 2 provides results related to the respective contribution of nested architecture, continual pre-training, and metadata extrapolation, while Table 4 provides a component-wise ablation of the SLIMP-based downstream classifier (image only, +lesion-level metadata, +patient-level metadata). We briefly summarize the main findings below.
>
> **Nested vs single-level contrastive loss:**
> In Table 2 we compare SLIMP with $SLIMP_{FLAT}$, a strong baseline that follows the standard contrastive learning approach, namely a single level InfoNCE loss between images and concatenated lesion/patient metadata, without the nested hierarchy. As can be seen in Table 2, SLIMP consistently outperforms $SLIMP_{FLAT}$, confirming the necessity of the nested hierarchy.
>
> **Effect of continual pre-training:**
> Table 2 also offers a comparison between Pre-SLIMP and $SLIMP_{IMAGE}$. The former employs the SLICE-3D pre-trained vision encoder, without continual pre-training on each target dataset, while $SLIMP_{IMAGE}$ adapts the vision encoder to each target dataset via multi-modal continual pre-training.  This comparison demonstrates the performance boost obtained by adapting the encoder to the target domain, steadily improving performance across four target datasets. Additionally, the comparison of these variants with the full SLIMP architecture which utilizes all three encoders (image, lesion-metadata, patient-metadata), further highlights the consistent performance improvement achieved from continual pre-training across the four target datasets.
>
> **Metadata Extrapolation:**
> Table 2 offers also a comparison between Pre-SLIMP, which is uni-modal (image only) and Ret-SLIMP that combines image features with features from metadata retrieved from SLICE-3D. Ret-SLIMP consistently improves performance over Pre-SLIMP and, in some cases, matches $SLIMP_{IMAGE}$, validating the utility of extrapolating metadata from the reference dataset. (Note: a similar discussion is provided to the answer of Q2 by reviewer SBYQ).
>
> **Component-wise ablation:**
> The impact of integrating modalities for the final downstream task is quantified in Table 4. The component-wise ablation clearly shows that starting from image-only features, recursively adding lesion-level metadata and then adding patient-level metadata **each contributes positively** to the overall classification performance, confirming that every component of the nested architecture adds informational value.

---

> ### Author Response · Authors · 2025-11-26
> **Answer to Reviewer CbJ2 (Part 3 - last part)**
>
> ### **Weakness 3/Question 2: Failure cases of metadata extrapolation**
>
> We thank the reviewer for raising this critical question regarding the potential risks and failure modes of the metadata extrapolation strategy (**Ret-SLIMP**). The concern about visually similar images having significantly different, clinically important metadata is highly valid and is addressed by analyzing the inherent noise introduced by this cross-domain retrieval.
>
> **Quantifying the extrapolation risk:**
> While a detailed clinical failure case analysis is constrained by the nature of medical data, due to privacy and re-identification concerns, we quantify the inherent error (risk) introduced by extrapolation by comparing performance metrics against a model using **true** metadata.
> Firstly, we observe from Table 2 that the performance of Ret-SLIMP, which use extrapolated metadata, is consistently lower than the full SLIMP model that uses true metadata. This performance gap reflects the noise and errors introduced by the retrieval process, effectively quantifying the risk that the model retrieves clinically divergent metadata. Nevertheless, despite this inherent noise, Ret-SLIMP achieves strong and competitive performance, often matching or surpassing the $\text{SLIMP}_{\text{IMAGE}}$ baseline which is adapted to the target dataset but uses no metadata. This stability, even with noisy pseudo-metadata, confirms that the retrieval process is robust enough for practical application.
>
> **Validation of Semantic Meaningful Retrieval:**
> The success of Ret-SLIMP relies on the high quality and structural preservation of the common embedding space learned during pre-training, minimizing the chance of arbitrary or catastrophic failures. In fact, Section E demonstrates that the embedding space is well-structured. Figure 4 confirms that, even under substantial domain shift (e.g., PAD-UFES-20), the distribution of similarity scores between target images and retrieved metadata remains aligned with the ground-truth distribution from the source dataset. Moreover, Table 7 provides concrete evidence of successful retrieval on the SLICE-3D validation set. The model retrieves the true metadata in the top-1 match over 45% of the time (R@1) and in the top-100 matches over 99% of the time, demonstrating that the image features accurately align with the corresponding tabular phenotype features.
>
> To qualitatively assess the semantic meaningfulness of the retrieval, we performed a visual analysis of the images corresponding to the top-retrieved metadata. We observed that the majority of images linked by the retrieval pipeline share similar visual and phenotypic characteristics (e.g., size, border irregularity and color), confirming that the embedding space successfully aligns features that clinicians consider relevant. We acknowledge that some instances show visually dissimilar lesions. These cases likely represent the failure modes where the model retrieves metadata based on underlying clinical variables (e.g., patient history) that override visual appearance, which accounts for the performance gap between Ret-SLIMP and the full SLIMP model. We will include a section in the Appendix showcasing these qualitative retrieval examples to further support the utility and limits of the Ret-SLIMP approach.
>
> **Architectural design mitigating misalignment:**
> As clarified in Figure 2 (right), our retrieval pipeline is sequential and multi-layered, first retrieving lesion-level metadata, then the resulting combined feature is aggregated to form an intermediate patient representation, and lastly this representation is used to retrieve the patient-level metadata. The lack of constraint between the retrieved lesion and patient metadata for the target domain forces the model to synthesize a likely patient **phenotype** before retrieving the macro-level patient data, making the retrieval process more robust to localized visual noise effectively mitigating simple one-to-one mapping errors.

---

### Official Review · Reviewer_b9ns · 2025-10-31

**Soundness:** 2
**Presentation:** 2
**Contribution:** 2
**Rating:** 4
**Confidence:** 4

**Summary:**

This paper proposes a contrastive learning module to align the skin disease image data with patients' meta data and diseases' meta data to achieve collaborative learning within all these elements. Through such a multimodal contrastive learning module, classification accuracy is higher and more robust.

**Strengths:**

1. The performance of the proposed model is better than existing methods due to the complementary information from metadata.
2. Sufficient ablation studies prove the effectiveness of the proposed modules.

**Weaknesses:**

1. The proposed method requests a visual image, disease meta data and patients meta data. So many information requirements will constrain the model generalization in different situations.
2. The proposed method introduces contrastive learning loss between image and tabular data. What is the difference from standard contrastive learning loss?
3. Some skin disease datasets are not similar to the used dataset. How does the model ensure the transferability of these datasets?
4. The evaluation mainly relies on benign simplification, and more fine-grained details are needed for real diagnosis

**Questions:**

1. How does the model ensure the smooth performance when patients' meta data is missing or noisy?
2. In the metadata extrapolation setting, how to deal with the situations if the reference and target datasets are very different.
3. Please address the difference with standard contrastive learning loss.

---

> ### Author Response · Authors · 2025-11-26
> **Answer to Reviewer b9ns (Part 1)**
>
> We sincerely thank the reviewer for their time and insightful feedback on our paper. We are pleased that the reviewer recognizes the strengths of SLIMP in achieving superior performance by leveraging complementary information from metadata and the effectiveness of our sufficient ablation studies. In the following, we attempt to address the reviewer’s concerns, questions and observations regarding our work.
>
> ### **Weakness 1: High information requirements may constraint model generalization**
>
> We thank the reviewer for raising this concern. While SLIMP's design is inspired by the holistic approach of clinicians, we explicitly designed it for flexibility and robust performance even when the required data modalities are partially unavailable or structurally divergent in the target domain. This is achieved via suitable mechanisms that fully exploit all the information provided by each target dataset, as described below.
>
> **Adapting to Different Metadata:**
> One of the main contributions of our work is the use of a continual pre-training strategy for allowing to adapt the pre-trained model to a target dataset with different metadata attributes than those considered in the SLICE-3D dataset. This efficiently adapts the metadata encoder to the new features by fine-tuning only the embedding layers, mitigating catastrophic forgetting.
>
> **Missing Patient Data:**
> For target datasets where patient-level metadata is completely unavailable, the model can be continually pre-trained and deployed using only the lesion-level contrastive loss (${L}_{lesion}$), successfully leveraging the remaining image and lesion metadata modalities.
>
> **Target Datasets Lacking All Metadata:**
> For the most constrained scenario where the target dataset contains no metadata, we introduce two variants, **$\text{Pre-SLIMP}$** and **$\text{Ret-SLIMP}$**. $\text{Pre-SLIMP}$ utilizes the image encoder produced via the multimodal pre-training on the SLICE-3D dataset, performing inference using only the images of the target dataset. $\text{Ret-SLIMP}$ on the other hand enriches the $\text{Pre-SLIMP}$ image features by extrapolating metadata from the reference dataset (SLICE-3D) based on the structure of the common embedding space. As shown in Table 2, $\text{Ret-SLIMP}$ achieved consistently improved classification performance over $\text{Pre-SLIMP}$ and comparable results to $\text{SLIMP}_{\text{IMAGE}}$, directly addressing the lack of metadata at the cost of slight performance.
>
> **Overall**, this multi-faceted approach demonstrates that SLIMP is designed to operate effectively across varying levels of data availability. Moreover, the **low-shot evaluation** in Table 2 shows that SLIMP's robust representations lead to remarkable label efficiency, outperforming most fully supervised baselines even when using only 10% of the labels. This is a crucial feature for real-world medical applications where data annotation is often expensive and time-consuming.

---

> ### Author Response · Authors · 2025-11-26
> **Answer to Reviewer b9ns (Part 2)**
>
> ### **Weakness 2/ Question 3: Novelty over standard InfoNCE formulation**
>
> [Answer reiterated also in: Reviewer CbJ2 (Part 1) and Reviewer 4Q9R (Part 1)]
>
> We agree with the reviewer that SLIMP uses the standard InfoNCE loss. The crucial difference and innovation lie not in modifying the loss formula but in the novel nested multi-modal scheme through which the standard loss is architecturally applied.
>
> **Nested architecture vs. flat loss:**
> Unlike standard contrastive learning, which operates on a single flat level (e.g., image–image or image–text pairs), or flatly combined modalities (as in $\text{SLIMP}_\text{FLAT}$), SLIMP introduces a two-level nested contrastive framework specifically designed to model the **compositional clinical hierarchy** of skin lesion data:
>
> - The **inner level** ($L_{lesions}$) maximizes the agreement between the lesion image ($w_p^l$) and the corresponding lesion-specific metadata ($h_l^p$) to obtain fine-grained lesion representations.
> - The **outer level** ($L_{patient}$) aggregates the lesion-level representations into a  patient phenotype summary ($z_p$) and then contrasts this summary with the patient-level metadata.
>
> This hierarchical and multi-modal structure explicitly encourages learning representations that incorporate patient-level context, leading to stronger, more generalizable features. To the best  of our knowledge, this structure is novel and fundamentally different from standard InfoNCE pipelines, which do not model nested, multi-modal clinical hierarchies.
>
> **Distinction from other hierarchical methods:**
> Moreover, the nested structure in SLIMP is fundamentally distinct from other hierarchical contrastive learning approaches. Compared to the works of Zhang et al. (2022) and Jiang et al. (2023) which enforce consistency across a **taxonomic or label hierarchy** (e.g., class lineage), SLIMP models a compositional hierarchy based on a relation of belonging (multiple lesions belong to a single patient). At the same time, our scheme spans multiple data modalities (image, lesion metadata, patient metadata) across its levels, contrasting with intra-modal hierarchical approaches like Wang et al. (2023), which apply contrastive loss across multiple levels of the same modality (as for example medical time-series).
>
> The effectiveness of this nested architecture is confirmed by our ablation study in Table 2. Collapsing our nested design into a single-level contrastive loss ($\text{SLIMP}_{\text{FLAT}}$) leads to substantial drops in performance (e.g., −4.9% BAcc loss on PAD-UFES-20 and −7.4% BAcc loss on HIBA) compared to the full nested SLIMP model. This clearly shows that the nested contrastive architecture, rather than a modified loss formula, is the crucial component driving SLIMP’s state-of-the-art gains.

---

> ### Author Response · Authors · 2025-11-26
> **Answer to Reviewer b9ns (Part 3)**
>
> ### **Weakness 3: Transferability to datasets dissimilar to SLICE-3D**
>
> **Continual pre-training adapts to divergent metadata and image domains:**
> Our multi-modal continual pre-training strategy is the core mechanism for ensuring transferability across domains that differ in metadata structure (e.g., attribute names, fields) and imaging modalities (clinical vs. dermoscopic). After pre-training on SLICE-3D which provides the most diverse and metadata-rich lesion–patient pairs, SLIMP **adapts to each target dataset by fine-tuning only the embedding layers** of both encoders to mitigate catastrophic forgetting of the rich representations learned on the large SLICE-3D dataset. As demonstrated in **Table 13**, full fine-tuning degrades performance, whereas embedding-only tuning consistently improves accuracy, balanced accuracy, and AUC across datasets.
>
> This selective adaptation mechanism is critical to ensuring smooth transferability from SLICE-3D to datasets that differ significantly in color distribution, acquisition modality, lesion appearance, and metadata structure. For instance, target datasets like HIBA and HAM10000 introduce different imaging modalities (Mixed/Dermoscopic) and varying clinical metadata attributes. Despite these differences, the representations obtained after continual pre-training ($\text{SLIMP}_{\text{IMAGE}}$ and SLIMP) significantly outperform competing baselines, demonstrating successful domain adaptation.
>
>
> **SLIMP representations are inherently robust and generalizable:**
> To directly test cross-dataset transferability, we compared SLIMP pre-trained on SLICE-3D with models pre-trained **independently on each target dataset** (Table 3). In all cases, the models trained exclusively on a target dataset, despite being evaluated on the same dataset, underperform relative to the SLIMP model pre-trained on SLICE-3D that undergoes continual pre-training. This confirms that SLIMP learns superior, more robust, phenotype-level representations on SLICE-3D that successfully generalize across heterogeneous domains.
>
> For datasets completely lacking metadata, our Ret-SLIMP extrapolation strategy successfully transfers knowledge by retrieving pseudo-modalities from SLICE-3D based on the quality of the learned embedding space. The resulting performance, which is comparable to $\text{SLIMP}_{\text{IMAGE}}$ (Table 2), shows that the learned features maintain discriminative power and structural alignment even when transferring to an unknown, metadata-free domain.
>
> ---
>
> ### **Weakness 4: Evaluation focused on benign vs. malignant**
>
> We acknowledge the reviewer's point regarding the need for fine-grained evaluation to ensure clinical relevance.
>
> **SLIMP addresses fine-grained diagnosis beyond binary classification:**
> Although our primary evaluation focuses on the binary benign-vs-malignant task to allow for consistent comparison across all target datasets with heterogeneous taxonomies and class imbalance, SLIMP is not limited to binary classification.
>
> As demonstrated in Appendix G.2 and summarized in Table 16, we perform full **multiclass classification** on the PAD-UFES-20 dataset. SLIMP substantially outperforms all competing baselines across all metrics in this more challenging, fine-grained lesion categorization task. We also introduce both Image-to-Metadata (I2T) and Metadata-to-Image (T2I) **retrieval tasks** (Tables 17–18), which operate independently of class labels altogether. These tasks require the model to capture subtle and nuanced clinical and phenotypic characteristics to achieve high-ranking performance, further supporting SLIMP’s use in fine-grained diagnosis.
>
> It is important to emphasize that **the entire pre-training stage uses no labels whatsoever**. SLIMP learns lesion phenotypes, reflecting the real clinical workflow where dermatologists assess risk based on holistic patient and lesion characteristics, not discrete labels. This is further supported by the **attention maps** (Figures 7–8), where SLIMP focuses on medically relevant lesion structures and morphological attributes far more consistently than baseline models,  suggesting superior feature learning for diagnostic tasks.

---

> ### Author Response · Authors · 2025-11-26
> **Answer to Reviewer b9ns (Part 4 - last part)**
>
> ### **Question 1: Handling missing, noisy or incomplete patient metadata**
>
> **SLIMP remains robust when patient metadata is missing, noisy, or incomplete:**
> We thank the reviewer for raising this important question. Our paper explicitly evaluates this scenario through both ablations and architectural design choices. **Table 4** presents a complete ablation study showing that SLIMP continues to perform strongly even when patient metadata, lesion metadata, or both are removed. Regarding missing patient metadata, the nesting mechanism provides a natural defense. In such scenarios, as in the case of PH2, the pre-training defaults to a variant using the lesion-level contrastive loss only ($\mathcal{L}_{lesions}$). This ensures that the model can be effectively utilized or fine-tuned even when operating exclusively on lesion images and their accompanying metadata.
> The robustness of SLIMP is further supported by employing the **TRACE** encoder. TRACE is specifically designed to handle heterogeneous clinical inputs containing missing fields, irregular values, and noise. The datasets we evaluate naturally exhibit these challenges: as summarized in the Table below (the fields of missing metadata will be added in Table 1 of the paper), real clinical datasets contain substantial missing metadata, for example, **PAD-UFES-20 has 32.2%** missing patient metadata, **HIBA has 21.2%**, and **PH2 contains no patient metadata at all (NA)** in its structure. Despite these real data constraints, SLIMP consistently exceeds the performance of the baselines across all datasets (Table 2). In addition to metadata robustness, SLIMP remains robust even when labels themselves are incomplete or scarce, which is a common constraint in medical domains. The low-shot results in Table 2 show that with only 10% or even 1% of downstream labels, SLIMP still outperforms most baselines trained with full labels. This demonstrates that SLIMP is resilient not only to missing metadata but also to incomplete supervision.
>
> | Datasets     | Patient Missing Values (%) | Lesion Missing Values (%) |
> |--------------|-----------------------------|----------------------------|
> | SLICE-3D     | 1.78                        | 0.04                       |
> | PAD-UFES-20  | 32.2                        | 7.00                       |
> | HIBA         | 21.2                        | 12.8                       |
> | HAM10000     | 0.57                        | 1.17                       |
> | PH2          | NA                          | 6.12                       |
>
> ---
>
> ### **Question 2: Metadata extrapolation under strong dataset shift**
>
> **SLIMP’s phenotype-driven embedding space generalizes across heterogeneous imaging and metadata domains:**
> We thank the reviewer for raising this point about how our approach handles large differences between the reference and target datasets. Section E provides a detailed analysis of the structure of the SLIMP embedding space under domain shift. There are three main types of domain shift to consider, namely shifts in the image modality, shifts in the metadata modality, or shifts in both. Our reference dataset (SLICE-3D) contains total-body photography images, whereas some target datasets differ substantially in image modality. For instance, **HIBA** consists primarily of dermoscopic images. Even under this large visual shift, **Ret-SLIMP improves Balanced Accuracy over Pre-SLIMP from 76.0%** to **81.3%**, and improves AUC from **0.845** to **0.861** (Table 2), clearly demonstrating that extrapolated metadata remain beneficial. Figure 4 further confirms that the retrieved metadata embeddings remain well aligned with the target image embeddings despite the modality gap. Another significant example is **PAD-UFES-20** where the domain shift is large in **both images** (smartphone photos rather than TBP or dermoscopy) and **metadata** (very few fields overlap with the SLICE-3D schema). Despite these differences, metadata extrapolation again yields clear improvements, with Balanced Accuracy increasing from **76.0% (Pre-SLIMP)** to **77.0% (Ret-SLIMP)**, and AUC increasing from **0.781** to **0.814**. Figure 4 shows also that even in this heterogeneous setting the similarity distributions between images and retrieved metadata remain close to the SLICE-3D distributions, indicating that the embedding geometry is preserved across domains. These quantitative results collectively indicate that SLIMP’s embedding space is robust to shifts in either modality, or both, and that metadata extrapolation remains effective even when the target dataset is very different from SLICE-3D.

---

### Official Review · Reviewer_SBYQ · 2025-11-04

**Soundness:** 1
**Presentation:** 3
**Contribution:** 2
**Rating:** 4
**Confidence:** 4

**Summary:**

The paper introduces SLIMP, a novel self-supervised pre-training framework for learning skin lesion representations by using a nested multi-modal contrastive learning on image-metadata pairs. The authors propose aligning lesion images with lesion-specific metadata and aggregating these representations to patient-specific images and metadata. The authors also propose a continual pre-training to adapt to other datasets and a retrieval-based method to extrapolate metadata in case of missing metadata. The proposed SLIMP outperformed existing methods on downstream classification tasks.

**Strengths:**

The paper proposed a well-motivated approach in contrastive pre-training for medical image-metadata pairs and offered practical solutions like continual pre-training and metadata extrapolation to adapt pre-trained methods to other datasets, even when the metadata is not available.

**Weaknesses:**

The SLICE-3D dataset has 393 malignant samples and around 400k benign samples. On the patient level, there are patients for whom there is no malignant sample, and even for patients with malignant samples, the number of malignant samples can be very small compared to the number of benign samples. Unlike supervised training that would have a detector specifically focus on the feature representing malignant lesions, pre-training on this dataset with only optimizing the distance between the image and metadata representation is expected to have difficulty teaching accurate representations to the model that can distinguish malignant samples.

Moreover, in their patient-level representation learning, they are average pooling over all the lesions of a patient, which is expected to further drown out the information from a malignant lesion over the information from a considerably higher number of benign lesions.

Their proposed positive sampling would work on the patients that actually have malignant lesions, which makes it less effective since there are generally more patients that have no malignant lesions in SLICE-3D and similar datasets.

The authors have mentioned that they have split the target datasets randomly for downstream tasks, which does not ensure there is no overlap between patients in the training and test sets and can potentially lead to overoptimistic performance metrics.

**Questions:**

1. Could the authors clarify more information regarding the downstream evaluation splits? It would strengthen the paper if the performance of SLIMP were shown on patient-disjoint training and testing sets.

2. In cases when a dataset has a distribution different from the SLICE-3D dataset (such as lesion types or patient demographics not well represented in SLICE-3D), would there be any performance benefit in retrieving the closest matching metadata?

---

> ### Author Response · Authors · 2025-11-26
> **Answer to Reviewer SBYQ (Part 1)**
>
> We sincerely thank the reviewer for the constructive feedback and for recognizing that we propose a **well-motivated** approach for contrastive pre-training on medical image-metadata pairs, and **practical solutions** such as continual pre-training and metadata extrapolation for adapting the model to diverse target datasets. We address the reviewer’s concerns and questions below.
>
> ### **Weakness 1: Concern about class imbalance in SLICE-3D undermining class-wise representations**
>
> **Pre-training objective of SLIMP framework:**
> We thank the reviewer for raising this point and appreciate the opportunity to clarify how the SLIMP pre-training objective operates, as this concern often arises from comparing self-supervised methods to supervised training techniques. In our framework, contrastive pre-training does not rely on malignant/benign labels and **does not attempt to learn class-discriminative boundaries during either the initial pre-training and the continual pre-training stages**. Instead, SLIMP optimizes instance-level alignment between each lesion image and its associated metadata. Because no class information is used in this process, the imbalance between malignant and benign samples does not influence the optimization objective. In practice, this makes SLICE-3D a well-suited dataset for self-supervised multimodal learning: it is large, diverse, and enriched with lesion- and patient-level metadata, even though it would be less appropriate for supervised training due to the imbalance reasons the reviewer notes. As a result, **the large number of benign samples contributes additional phenotypic and contextual variability**, which in turn supports more robust and transferable representation learning.
>
> **Self-supervised methods comparison:**
> The representations learned under this class-agnostic objective demonstrably acquire robustness. As shown in Table 2, when comparing the features extracted from our image encoder alone (Pre-SLIMP) against features from purely image-based self-supervised methods like SimCLR and supervised imbalance-handling methods like SBCL, we observe that Pre-SLIMP achieves similar performance to the former and often outperforms the latter on downstream tasks. This finding confirms that the metadata signal is successfully encoded into the visual representation, resulting in features that are inherently robust and discriminative, even without direct supervision on the rare malignant class.
>
> **In-domain pre-training:**
> We would also like to highlight that the reviewer’s concern is directly tested through our ablation in Table 3 (p.8) of the submitted manuscript. Here, we perform in-domain pre-training on datasets that contain a much more balanced malignant-to-benign ratio (PAD-UFES-20, HIBA, HAM10000). The results in Table 3 show that models trained from scratch on these better balanced but smaller datasets consistently underperform the model pre-trained on the SLICE-3D dataset. This hierarchy confirms that dataset scale and richness outweigh class balance for the self-supervised objective.
>
> **Out-of-domain pre-training with HAM10000:**
> We also conducted out-of-domain pretraining on the HAM10000 dataset, which contains 10,015 dermoscopic images and is significantly larger than PAD-UFES-20 (\~2.3K images) and HIBA (\~1.6K images) and far more balanced in terms of malignant vs. benign distribution than SLICE-3D (\~1/5 instead of \~1/1000). Using the same evaluation pipeline as in our main experiments, we perform downstream classification on both PAD-UFES-20 and HIBA achieving 89.6% and 90.7% accuracy respectively, still underperforming $\text{SLIMP}_{B=4}$.
>
> **In Summary: SLIMP (SLICE-3D) > SLIMP (HAM10000) > SLIMP (in-domain PAD/HIBA).**
> This hierarchy is directly analogous to dataset scale and metadata completeness (table below). This pattern strongly supports our conclusion that **multimodal dataset richness and size are far more important than malignant/benign balance for self-supervised contrastive learning**.
>
> | Datasets     | Patient Missing Values (%) | Lesion Missing Values (%) |
> |--------------|-----------------------------|----------------------------|
> | SLICE-3D     | 1.78                        | 0.04                       |
> | PAD-UFES-20  | 32.2                        | 7.00                       |
> | HIBA         | 21.2                        | 12.8                       |
> | HAM10000     | 0.57                        | 1.17                       |
> | PH2          | NA                          | 6.12                       |

---

> ### Author Response · Authors · 2025-11-26
> **Answer to Reviewer SBYQ (Part 2)**
>
> ### **Weakness 2: Average pooling over all lesions per patient will drown out malignant-lesion information**
>
> In continuation of the previous answer, we thank the reviewer for raising this point, giving us the opportunity to clarify a misunderstanding of our pre-training objective, which is not to detect malignancy directly but to learn class-agnostic, holistic patient phenotypes from multimodal lesion and patient information.
>
> **Individual lesion contribution:**
> Malignant lesions **are not “averaged away” before they are learned**. At the lesion level (inner), the inner InfoNCE loss ($L_{lesions}$) maximizes agreement between the lesion image features ($w_p^l$) and lesion metadata features ($h_p^l$) for every individual lesion. This ensures that the unique visual and clinical characteristics of a specific malignant lesion are encoded into a robust, rich multimodal representation $(w_p^l, h_p^l)$, with $ l=1 \ldots N_p $, **before the averaging step**. Only afterwards are these combined lesion representations average-pooled into a single vector that summarizes the patient’s lesion phenotype, which is then aligned with the patient-level metadata via the outer InfoNCE loss ($L_{patient}$). This design ensures that lesion-specific cues (including those from a malignant lesion) directly influence the decision, while the patient phenotype provides complementary context rather than overwhelming lesion-level information.
>
> Overall, the superiority of the proposed nested architecture is validated empirically, where our SLIMP model consistently and significantly outperforms the single-level (flat) variant $\text{SLIMP}_{\text{FLAT}}$ across all datasets (Table 2). This demonstrates that the nested contrastive learning successfully preserves the essential lesion-level information while leveraging the patient context.
>
> ---
>
> ### **Weakness 3: Positive sampling is ineffective because most patients have no malignant lesions**
>
> **“Positive sampling” mechanism motivation:**
> In continuation of the previous answers, we should highlight that our positive sampling strategy is **not designed as a malignancy-focused mechanism**, but as a **diversity-aware lesion sampler**. SLIMP’s pre-training remains fully class-agnostic. The goal of positive sampling is to ensure that, when we subsample N lesions per patient, we capture more heterogeneous and informative lesion–metadata pairs (including rare or atypical lesions).
>
> **Patient-level imbalance:**
> The sampling is relevant because, at the patient level, the data distribution is substantially less severe than at the lesion level (99.9% benign lesions). We consider a patient with malignancy any patient with at least one malignant lesion. Thus, a non-negligible fraction of the 1,042 total patients  contribute malignant information. Positive sampling helps ensure that these informative lesions are consistently included among the sampled lesions for those patients, maximizing exposure to rare, high-value data points during the multi-modal alignment.
>
> **Uniform vs. Positive Sampling:**
> We trained a SLIMP variant on SLICE-3D with uniform lesion sampling (w/o p.s) and compared it against our default positive sampling configuration (w/ p.s). As reported in the table below, positive sampling provides consistent, though marginal, performance gains. For example, on the HIBA dataset, positive sampling improves the Balanced Accuracy (BAcc) from 89.3 to 91.9 and AUC from 0.933 to 0.954. This result confirms that our conclusions do not depend critically on this design choice, but that the heuristic aids in capturing diverse, high-value lesion characteristics, slightly improving the resulting representations.
>
> |             | **PAD** |      |      | **HIBA** |      |      | **HAM** |      |      |
> |-------------|---------|------|------|----------|------|------|---------|------|------|
> |             | Acc     | BAcc | AUC  | Acc      | BAcc | AUC  | Acc     | BAcc | AUC  |
> | **w/o p.s** | 89.6    | 89.9 | .926 | 89.5     | 89.3 | .933 | 86.0    | 84.6 | .919 |
> | **w/ p.s**  | 90.9    | 90.2 | .926 | 92.0     | 91.9 | .954 | 87.3    | 83.5 | .923 |

---

> ### Author Response · Authors · 2025-11-26
> **Answer to Reviewer SBYQ (Part 3)**
>
> ### **Weakness 4/Question 1: Possible overoptimistic performance due to non–patient-disjoint downstream splits**
>
> We thank the reviewer for raising this point, as enforcing patient-disjoint splits is critical for verifying generalization in clinical settings.
>
> **Standard evaluation Protocol:**
> We would like to clarify that our evaluation protocol, in general, follows the standard practice established in the self-supervised learning (SSL) literature. In prominent SSL methods such as SimCLR  (Chen et al., 2020), MAE (He et al., 2022), and DINOv2 (Oquab et al., 2023), the self-supervised pre-training is performed on the ImageNet-1k training split, and the downstream supervised evaluation (linear probe or fine-tuning) is also typically trained on the ImageNet-1k training split. Similarly, our lesion-level splits follow this principle for evaluating the intrinsic quality of the feature representations. Specifically, our standard downstream evaluation protocol uses random lesion-level splits (90%-10% split ratio) which is a common practice for evaluating features learned via self-supervised pre-training (e.g., linear probing on frozen features).
>
> **Patient disjoint experiments:**
> To directly address the reviewer’s suggestion, assessing whether our performance metrics are not overoptimistic due to potential patient overlap, we additionally repeated all downstream linear probing experiments using strictly patient-disjoint splits. In these splits we ensure that all lesions from a given patient are assigned to the same split, while keeping an overall split ratio as close as possible to 90%/10%, as far as lesions are concerned. The resulting metrics are reported below.
>
> | Dataset (disjoint splits) | Acc | BAcc | F1 | AUC |
> |--------|-----|------|-----|-----|
> | **PAD-UFES-20** | 89.5% | 89.6% | .894 | .940 |
> | **HIBA** | 91.7% | 91.8% | .915 | .954 |
> | **HAM10000** | 87.9% | 80.8% | .676 | .926 |
>
> | Dataset (multiple seeds) | Acc (mean±std) | BAcc (mean±std) | F1 (mean±std) | AUC (mean±std) |
> |--------|------------------|------------------|----------------|----------------|
> | **PAD-UFES-20** | 90.6% ± 0.9% | 90.4% ± 0.8% | .909 ± .017 | .935 ± .001 |
> | **HIBA** | 91.2% ± 1.4% | 91.2% ± 1.4% | .911 ± .016 | .951 ± .010 |
> | **HAM10000** | 87.8% ± 0.4% | 81.6% ± 1.7% | .694 ± .016 | .926 ± .0003 |
>
> In the **top three rows**, we report the performance obtained when we instead enforce **patient-disjoint splits**. In the **bottom three rows**, we report the downstream performance of SLIMP using our **standard lesion-level** splitting strategy, evaluated over three random seeds, and we report the corresponding mean and standard deviation of each metric. We observe that the patient-disjoint results consistently fall within the mean ± standard deviation of the metrics obtained with our standard lesion-level splits across three target datasets. For instance, the Balanced Accuracy for HIBA with disjoint splits is 91.8, which is well within the range of our multi-seed, lesion-level splits (91.2 ± 1.4). In several cases (HIBA, HAM10000), the patient-disjoint results even outperform the average of the lesion-level splits. In the revised version of the manuscript we will explicitly clarify the splitting strategy to avoid misunderstandings and will include the patient-disjoint experiment as an ablation to further strengthen the paper.

---

> ### Author Response · Authors · 2025-11-26
> **Answer to Reviewer SBYQ (Part 4 - last part)**
>
> ### **Question 2: Benefit of metadata retrieval under domain-shifts**
>
> **Pre-SLIMP vs Ret-SLIMP:**
> Our experiments with $\text{Pre-SLIMP}$, $\text{Ret-SLIMP}$, and $\text{SLIMP}_{\text{IMAGE}}$ (Table 2) directly address the situation where the target dataset has a different distribution from SLICE-3D and, more importantly, where metadata on the target domain are absent or not used.
> - **$\text{Pre-SLIMP}$** uses only the image encoder pre-trained on SLICE-3D (no target metadata and no continual learning)
> - **$\text{Ret-SLIMP}$** uses only the image encoder pre-trained on SLICE-3D but augments each target image with retrieved lesion- and patient-level metadata from SLICE-3D (reference metadata and no continual learning)
> - **$\text{SLIMP}_{\text{IMAGE}}$** performs continual multi-modal pre-training on the target dataset but uses only the adapted image encoder at downstream time (no metadata in the classifier)
>
> Table 2 shows that, across PAD-UFES-20, HIBA and PH2, $\text{Ret-SLIMP}$ consistently improves over $\text{Pre-SLIMP}$, demonstrating that retrieving the closest metadata is beneficial even when the target dataset differs from SLICE-3D in lesion types and demographics. For example on HIBA, $\text{Ret-SLIMP}$ improves the accuracy of $\text{Pre-SLIMP}$ by 5.6% (from 75.9% to 81.5%) and similarly on PAD-UFES-20 by 0.5% (from 76.5% to 77.0%). In several cases $\text{Ret-SLIMP}$ is competitive with, or even surpasses, $\text{SLIMP}_{\text{IMAGE}}$, despite never seeing target-dataset metadata, highlighting the strength of the learned cross-modal alignment.
>
> **Embedding space structure under domain-shifts:**
> Section E within the Appendix, further supports the validity of this retrieval procedure. On SLICE-3D, matching image-metadata pairs are well separated from non-matching ones in cosine similarity space (Table 6), showcasing a well-structured embedding space. Figure 4 then shows that, for target datasets such as HAM10000, PAD-UFES-20, and HIBA, the similarity distributions between the **target images** and **retrieved metadata** from SLICE-3D, although shifted to lower values (as expected under domain shift), remain tightly concentrated with reasonably high medians. This confirms that retrieval finds semantically meaningful metadata out of domain, explaining the performance gains.

---

### Author Response · Authors · 2025-12-03
**Summary of discussion and changes**

We thank all reviewers for highlighting the strengths of our work and for the constructive comments. We briefly summarize our core responses regarding Novelty and Performance below.

1. **Fundamental Architectural Innovation:** Reviewers **b9ns**, **CbJ2** and **4Q9R** raised concerns regarding the novelty of our approach. We respectfully disagree and we would like to clarify that SLIMP moves beyond standard contrastive approaches by introducing a **nested multi-modal contrastive framework** that explicitly models the clinical hierarchy (“multiple lesions → one patient”) across image, lesion metadata, and patient metadata. This architectural design is a **structural necessity** for correctly modeling medical data, producing representations that incorporate patient-level context and yield stronger, more generalizable features than flat InfoNCE pipelines. We empirically demonstrate that collapsing this nested design into a single-level contrastive baseline (**SLIMP_FLAT**) results in significant performance degradation (e.g., **–7.4% Balanced Accuracy on HIBA**), confirming that the nested architecture is the primary driver of our performance gains.

2. **Reliability of metadata extrapolation:** Reviewers **b9ns**, **CbJ2** and **4Q9R** questioned the validation and potential risks of the retrieval-based extrapolation (Ret-SLIMP). Regarding validation, in Section E we demonstrate high retrieval quality on the SLICE-3D validation set (**99.1% Recall@100**), confirming semantic alignment. As far as robustness is concerned, Figure 4 shows that embedding alignment is preserved even under large domain shifts (e.g., smartphone images in PAD-UFES-20). Crucially, Ret-SLIMP consistently outperforms Pre-SLIMP (image-only) across all datasets (Table 2). This proves that while extrapolation introduces noise, it does **not** introduce harmful bias. Instead, it provides informative context that yields a measurable performance gain.

3. **Class-agnostic objective and use of SLICE-3D:** Additionally, SLIMP’s pre-training objective is **class-agnostic in both initial and continual pre-training stages**. It does not attempt to learn malignant vs. benign decision boundaries but to learn rich multimodal patient and lesion representations. This justifies the use of SLICE-3D as a reference dataset. While the high benign count of the dataset might hinder supervised learning, for SLIMP it contributes crucial phenotypic and contextual variability. This design directly alleviates two major constraints of the medical domain: (a) **limited annotations**, and (b) **severe class imbalance**, as evidenced by our superior low-shot performance (Table 2).

4. **Substantial and consistent improvements:** The performance gains of SLIMP are robust and substantial when compared to relevant baselines. Importantly, SLIMP outperforms **SimCLR** (pre-trained on the same SLICE-3D data) by **+14.0%** in Average Balanced Accuracy (91.9% vs 77.9%), proving the value of the methodology beyond data exposure. Moreover, SLIMP (ViT-Small) outperforms the strongest domain-specific foundation model, **WL-CLIP (ViT-Large)**, by **+6.7%** in Average Balanced Accuracy. It also surpasses the dermatology foundation model **PanDerm-Large (Yan et al., 2025)** (added after suggestion of reviewer 4Q9R) by **+10.4% on PAD-UFES-20** and **+4.7% on HIBA**. Finally, the strength of the learned representations allows SLIMP to surpass fully supervised baselines even when using only **10% of the available labels**, highlighting its practical superiority for label-scarce medical applications.

5. **Rigorous evaluation protocol:** Reviewer **SBYQ** raised concerns regarding potential data leakage due to random splits. While our original protocol followed standard SSL benchmarks (random lesion-level splits), we addressed this by re-evaluating SLIMP using **strictly patient-disjoint splits**. The results consistently fall within the statistical margin of the original runs (e.g., HIBA Balanced Accuracy: **91.8% disjoint vs. 91.2% ± 1.4% random**). This confirms that our substantial performance gains are driven by generalizable feature learning, not patient overlap or overfitting.

We believe these clarifications fully address the concerns regarding the significance and validity of our contribution.

---

### Author Response · Authors · 2025-12-03
**Submission of revised paper**

We have submitted a revised version of the manuscript. The main changes are highlighted in blue in the manuscript and summarized here:

- Updated the structure and presentation of Section 3 to improve methodology clarity, providing additional details in Section D and ablations in Section F.
- Added a new ablation in Section F of Supplementary Material, comparing uniform and positive lesion sampling during SLIMP pre-training on SLICE-3D.
- Added a new ablation in Section F of Supplementary Material with strictly patient-disjoint splits for all downstream linear probing experiments.
- Extended Table 2 (sub-section 4.4) with PanDerm-Base and PanDerm-Large.
- Updated Table 1 (sub-section 4.1) to include lesion-level and patient-level missing metadata percentages for each dataset.
- Expanded Section D of Supplementary Material to explicitly detail the loss functions used at every stage of the pipeline: InfoNCE for pre-training on SLICE-3D, InfoNCE for continual pre-training on target datasets, and cross-entropy for downstream supervised classification.

Finally, we will make use of the 10th page of the camera-ready version to move the details regarding the structure of the embedding space reported in Section E to the main paper.

We thank the reviewers again for their comments and remarks that helped make the paper even stronger and more clear, and we express our regret for not being able to continue the constructive discussion with them due to the circumstances.

---

### Meta-Review · Area_Chair_N2Rt · 2026-01-06

**Summary:**

This paper aims to address multi-modal skin disease diagnosis by proposing a two-level contrastive learning framework. The paper originally received mixed reviews, with key concerns focusing on limited novelty and insufficient experimental details.

In general, the proposed two-level design appears somewhat trivial for this task, as it remains difficult to understand what complementary information each modality contributes. Clarifying the specific role and added value of each modality is a crucial aspect of multi-modal diagnosis, yet the paper does not provide adequate analysis in this regard. Consequently, it is hard to interpret how particular image features or lesions lead to the final diagnosis, and the method offers limited explainability concerning which visual patterns drive the predictions.

**Reviewer Concerns:**

The experimental details are well addressed, but the model novelty is not, even without more interpretable results.

**Reviewer Scores:**

More motivation about the model design for multi-modal data fusion. More intuitive interpretable results

---

### Decision · Program_Chairs · 2026-01-26

Reject